# Effects of early-life antibiotics on the developing infant gut microbiome and resistome: a randomized trial

Marta Reyman [1,2], Marlies A. van Houten[2], Rebecca L. Watson[3], Mei Ling J. N. Chu[1], Kayleigh Arp[1], Wouter J. de Waal[4], Irene Schiering[2], Frans B. Plötz [5], Rob J. L. Willems[6], Willem van Schaik [7], Elisabeth A. M. Sanders [1,8] & Debby Bogaert [1,3 ✉]

Broad-spectrum antibiotics for suspected early-onset neonatal sepsis (sEONS) may have pronounced effects on gut microbiome development and selection of antimicrobial resistance when administered in the first week of life, during the assembly phase of the neonatal microbiome. Here, 147 infants born at ≥36 weeks of gestational age, requiring broad-spectrum antibiotics for treatment of sEONS in their first week of life were randomized 1:1:1 to receive three commonly prescribed intravenous antibiotic combinations, namely penicillin + gentamicin, co-amoxiclav + gentamicin or amoxicillin + cefotaxime (ZEBRA study, Trial Register NL4882). Average antibiotic treatment duration was 48 hours. A subset of 80 non-antibiotic treated infants from a healthy birth cohort served as controls (MUIS study, Trial Register NL3821). Rectal swabs and/or faeces were collected before and immediately after treatment, and at 1, 4 and 12 months of life. Microbiota were characterized by 16S rRNA-based sequencing and a panel of 31 antimicrobial resistance genes was tested using targeted qPCR. Confirmatory shotgun metagenomic sequencing was executed on a subset of samples. The overall gut microbial community composition and antimicrobial resistance gene profile majorly shift directly following treatment ($R^2 = 9.5\%$, adjusted $p$-value = 0.001 and $R^2 = 7.5\%$, adjusted $p$-value = 0.001, respectively) and normalize over 12 months ($R^2 = 1.1\%$, adjusted $p$-value = 0.03 and $R^2 = 0.6\%$, adjusted $p$-value = 0.23, respectively). We find a decreased abundance of *Bifidobacterium* spp. and increased abundance of *Klebsiella* and *Enterococcus* spp. in the antibiotic treated infants compared to controls. Amoxicillin + cefotaxime shows the largest effects on both microbial community composition and anti-microbial resistance gene profile, whereas penicillin + gentamicin exhibits the least effects. These data suggest that the choice of empirical antibiotics is relevant for adverse ecological side-effects.

[1] Department of Pediatric Immunology and Infectious Diseases, Wilhelmina Children's Hospital and University Medical Center Utrecht, Utrecht, the Netherlands. [2] Department of Pediatrics, Spaarne Gasthuis, Hoofddorp and Haarlem, the Netherlands. [3] Medical Research Council and University of Edinburgh Centre for Inflammation Research, Queen's Medical Research Institute, University of Edinburgh, Edinburgh, UK. [4] Department of Pediatrics, Diakonessenhuis, Utrecht, the Netherlands. [5] Department of Pediatrics, Tergooiziekenhuis, Blaricum, the Netherlands. [6] Department of Medical Microbiology, University Medical Centre Utrecht, Utrecht, the Netherlands. [7] Institute of Microbiology and Infection, University of Birmingham, Birmingham, UK. [8] National Institute for Public Health and the Environment, Bilthoven, the Netherlands. ✉email: d.bogaert@ed.ac.uk

The importance of the human gut microbiome in health and disease is becoming increasingly clear. Disturbances of the gut microbial community composition after birth are associated with a broad scale of health problems in early infancy and later in life, such as infantile colic, wheezing, allergies, functional gastrointestinal disorders, obesity and generally an altered immune development[1–6]. The various causes of disturbances are also becoming evident. Among others, the effects of caesarean section (CS) delivery, formula feeding (as opposed to breastfeeding) and antibiotics on the developing neonatal gut microbiome have been described[7,8]. Antibiotic treatment, in particular, alters species diversity (α-diversity) and community composition, i.e. the ecology, of the gut microbiome for a prolonged period of time in both children and adults[9,10]. Antibiotic-induced shifts in gut microbiome composition subsequently result in the selection of antimicrobial resistance (AMR)[11]. The ecological side effects of antibiotics may be even more pronounced and persistent when administered in the early assembly phase of the neonatal gut microbiome in the first weeks of life. However, in this phase of life, broad-spectrum antibiotics are prescribed in up to 10% of all neonates because of (suspected) early-onset neonatal sepsis (sEONS)[12,13].

The most common pathogens causing EONS are group B streptococcus, *Listeria monocytogenes* and *Escherichia coli*. These organisms are susceptible to antibiotic regimens involving combinations of penicillin with gentamicin or amoxicillin combined with cefotaxime, making these treatments equally effective[14]. However, already two decades ago, a Dutch study performed on a neonatology intensive care unit highlighted the issue of amoxicillin driving the overgrowth of β-lactamase producing bacteria, such as *Klebsiella* species (spp.)[15]. Another concern raised was that third-generation cephalosporins, such as cefotaxime, select for resistant *Enterobacter* spp. strains[15]. The full ecological impacts of empiric antibiotic treatment administered in early-life on microbiome and AMR gene composition are still not well-known.

To identify the antibiotic regimen with the least ecological and AMR gene selection effects, we performed the ZEBRA study in which we enrolled 147 infants born at term, either by natural delivery or by secondary CS (SCS), for whom broad-spectrum antibiotics were indicated in the first week of life because of sEONS, and randomized them over three most commonly prescribed intravenous antibiotic combinations for this indication in the Netherlands, namely penicillin + gentamicin, co-amoxiclav + gentamicin or amoxicillin + cefotaxime[16]. On average, the antibiotic treatment duration was 48 h. In this work, we find that antibiotic-treated infants show temporarily reduced gut microbial diversity, and major and prolonged ecological perturbations detected using 16S rRNA-based sequencing, compared with healthy term-born controls, as well as shifts in AMR gene profile, as determined by quantitative polymerase chain reaction (qPCR) of a panel of 31 clinically relevant AMR genes. The compositional and AMR gene findings are confirmed by metagenomic shotgun sequencing (MGS) of a subset of samples. Also, we find marked differences in ecological perturbation between the three different antimicrobial regimens, suggesting that, next to adequate treatment, the choice of empirical antibiotics is also relevant for adverse ecological side-effects.

## Results
**Population characteristics**. The development of microbiome and resistome was studied in 147 neonates born ≥36 weeks of gestational age, who were recruited in the period from 16 January 2015 to 13 September 2016 in three different Dutch hospitals, and were followed until their first birthday. Neonates were randomized over the following antibiotic regimens (49 per group): penicillin + gentamicin, co-amoxiclav + gentamicin or amoxicillin + cefotaxime. As controls, 80 age-matched term-born infants, born between 19 December 2012 and 2 November 2014 were included from a previous healthy birth cohort study conducted in two of the same hospitals[17]. All infants were born in hospital except for three spontaneous home births in the antibiotic-treated group (one in each regimen) and five in the control group. From two infants in the amoxicillin + cefotaxime group, no samples were available for analysis, so these infants were excluded from further analyses (Supplementary Fig. 1). Eight (5.4%) children were lost to follow-up due to parents experiencing the collection of samples or completion of questionnaires as too burdensome, or moving abroad.

In line with sEONS characteristics, neonates treated with antibiotics were more often born with SCS, had a slightly shorter gestational age, a lower Apgar score at 5 min, and their mothers had on average a longer period of ruptured membranes before delivery (Supplementary Table 1). Antibiotic treated neonates were more often the first child of parents. As a consequence of the higher rate of SCS and sEONS, the antibiotic-treated neonates were hospitalized for a longer period of time. Antibiotic treatment after the first week of life was uncommon (43 out of all 225 participants), administered relatively late in the first year (average age 8 months), and did not differ between sEONS infants and controls at baseline. As expected, baseline characteristics did not differ between the infants randomized to each of the three antibiotic regimens, except for antepartum maternal antibiotics exposure. In total, only two of the sEONS infants showed to have a culture-proven sepsis.

**Effects of early-life broad-spectrum antibiotics on infant gut microbial composition development**. Antibiotic treatment led to a decrease in α-diversity (median Shannon index 0.77 before, versus 0.59 after antibiotic treatment in the sEONS infants, Wilcoxon test, $p = 0.01$), which was most outspoken immediately after antibiotic treatment (median Shannon index 0.59 in treated infants, versus 1.21 in controls at time point 2, Wilcoxon test, $p < 0.001$; Supplementary Fig. 2). Following, the α-diversity gradually recovered, though remained significantly lower throughout the first year of life (linear mixed model, $p = 0.01$). When stratified, we observed that the co-amoxiclav + gentamicin group had the lowest α-diversity immediately after treatment (median Shannon index 0.43; Fig. 1) which persisted throughout the first year of life when compared with controls (linear mixed model, $p = 0.002$). In inter-regimen comparisons, no significant difference in α-diversity was observed.

Before start of antibiotics, the overall microbial community composition did not significantly differ between neonates with sEONS and controls (permutational analysis of variance [PERMANOVA]-test, $R^2 = 1.2\%$, $p.\text{adj} = 0.14$; Fig. 2a), however, a large shift in composition was observed directly following treatment ($R^2 = 9.5\%$, $p.\text{adj} = 0.001$; Fig. 2b). Despite gradual recovery over time, a small but significant difference in composition remained at 12 months of age ($R^2 = 1.1\%$, $p.\text{adj} = 0.03$, Fig. 2c). The largest effect was found for the amoxicillin + cefotaxime group immediately after treatment ($R^2 = 14.7\%$, $p.\text{adj} = 0.003$; Fig. 2e), which was also significantly different from the other two regimens (versus co-amoxiclav + gentamicin: $R^2 = 7.6\%$, $p.\text{adj} = 0.003$; versus penicillin + gentamicin: $R^2 = 4.9\%$, $p.\text{adj} = 0.004$). Also, at age 1 month, the difference in composition for the amoxicillin + cefotaxime group versus controls was most outspoken, though subsequently, similar recovery patterns were observed as for the other two regimens (Fig. 2f). To exclude potential confounding effects of antibiotic

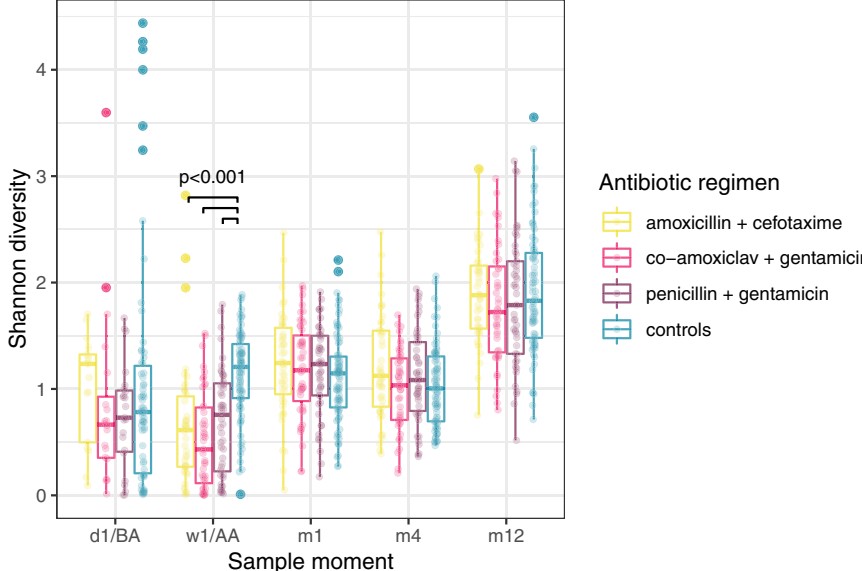

**Fig. 1 α-diversity.** Differences in Shannon diversity of the faecal microbiota between infants treated with the three studied antibiotic regimens and controls plotted per time point. Group differences were calculated using one-sided Wilcoxon tests. Boxplots with medians are shown; the lower and upper hinges correspond to the first and third quartiles (the 25th and 75th percentiles); the upper and lower whiskers extend from the hinge to the largest and smallest value no further than 1.5*IQR from the hinge; outliers are plotted individually by opaque circles; translucent circles visualize all data points (yellow = amoxicillin + cefotaxime [$n = 17$, $n = 46$, $n = 46$, $n = 45$ and $n = 44$ for the consecutive time points], pink = co-amoxiclav + gentamicin [$n = 20$, $n = 45$, $n = 47$, $n = 46$ and $n = 44$], purple = penicillin + gentamicin [$n = 23$, $n = 48$, $n = 47$, $n = 46$ and $n = 45$], teal = controls [$n = 66$, $n = 80$, $n = 80$, $n = 78$ and $n = 74$]. D = day, w = week, BA = before antibiotics, AA = after antibiotics, m = month. We observed a significant difference in Shannon diversity between each antibiotic regimen and the controls directly after antibiotic treatment ($p = 4.915e-09$ for amoxicillin + cefotaxime vs. controls, $p = 4.056e-10$ for co-amoxiclav + gentamicin vs. controls and $p = 9.128e-07$ for penicillin + gentamicin vs. controls). Source data are provided as a Source Data file.

treatment during follow-up (after the first week of life), we performed a post-hoc analysis on the subset of sEONS and control infants who had not received antibiotics during follow-up ($n = 123$ and $n = 59$, respectively), showing similar results with even larger effect sizes compared to the primary comparison, underwriting that initial antibiotic treatment was explaining the differences found between groups (Supplementary Table 2).

The stability of microbial community development within individuals over time was lower in antibiotic-treated infants compared to controls during the first 4 months of life (Supplementary Fig. 3). Again, this was most pronounced for infants in the amoxicillin + cefotaxime group, with a significant difference compared to both the control group and the penicillin + gentamicin group (after antibiotics vs. month 1 comparison, Wilcoxon test, $p < 0.001$ and $p = 0.001$, respectively).

**Clinical covariates associated with infant gut microbial composition development**. Besides antibiotic treatment ($R^2$ 1.4%), covariates significantly associated with overall gut microbial community composition in the overarching cohort were: age ($R^2$ 4.1%), day care attendance ($R^2$ 1.0%), breastfeeding at the time of sampling ($R^2$ 0.7%) and mode of delivery ($R^2$ 0.1%, all $p$.adj = 0.001). Covariates that differed at baseline between the antibiotic-treated infants and controls were: gravidity of mothers, gestational age, duration of ruptured membranes, duration of hospital stay after birth, presence of siblings <5 years of age and inhouse smoking. As of these variables mode of delivery was the only variable that was associated with microbial community composition over the first year of life, this variable was corrected for in all downstream microbiota analyses.

No significant association was found between overall microbial community composition and antibiotic treatment duration in the first week of life nor with maternal antepartum antibiotics when analyzed per time point (cross-sectionally). In a temporal,

multivariable analysis (sEONS group only), we found that both variables were, although significantly, only modestly associated with composition (antibiotic treatment duration $R^2$ 0.3%; maternal antepartum antibiotics $R^2$ 0.1%, both $p$.adj < 0.001), when compared with antibiotic regimen ($R^2$ 0.9%), age ($R^2$ 5.2%), day care attendance ($R^2$ 1.0%), and breastfeeding at time of sampling ($R^2$ 0.8%, all $p$.adj < 0.001). Given only two infants showed a culture-proven sepsis, further sub-analyses for this variable were not possible. However, to account for the likelihood of neonatal sepsis in our analyses, we further categorized our treated group according to the duration of antibiotic treatment into short (1–4 days) and long (>4 days) and studied the effect on overall microbial community composition in a cross-sectional manner (Supplementary Table 3), showing no significant difference at any time point between the infants receiving short or long antibiotic treatment. When we repeated the temporal, multivariable analysis including the categorized (instead of the continuous) antibiotic duration variable, the effect size on overall microbiota composition was still modest ($R^2$ 0.4%).

As maternal antepartum antibiotics were the only variable that differed between the sEONS groups at baseline and was associated with microbial community composition over the first year of life, this variable was corrected for in all downstream inter-regimen microbiota analyses.

**Differences in microbial colonization patterns between antibiotic-treated infants and controls**. The colonization patterns in the control group over time were similar to those described in previous healthy infant studies (Fig. 3)[7,18]. Facultative anaerobic genera such as *Escherichia*, and *Staphylococcus* were highly abundant in the first samples collected directly after birth, rapidly followed by the predominance of the genus *Bifidobacterium*. Next, temporal analysis showed a total of 251 differentially abundant Operational Taxonomical Units (OTUs) between

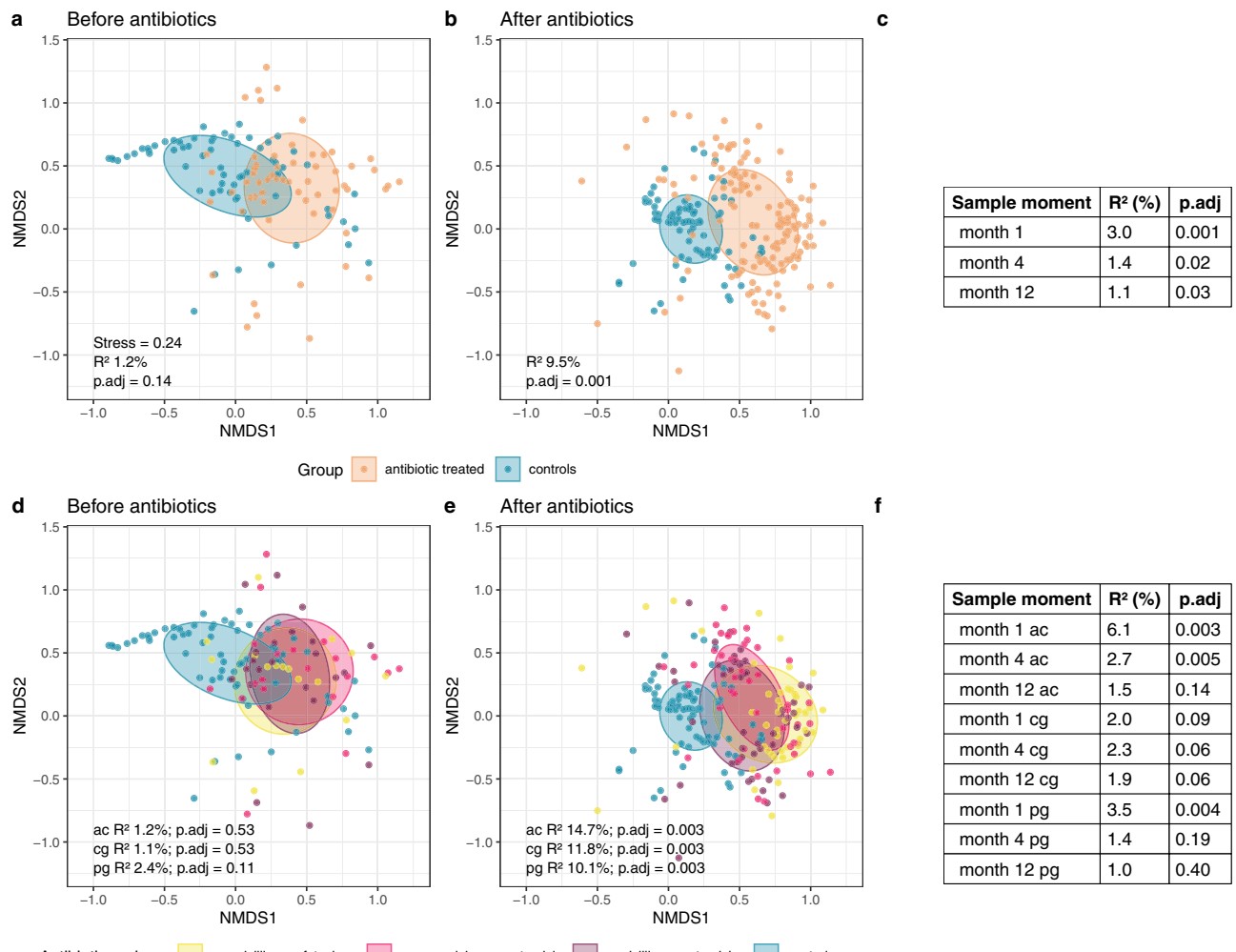

**Fig. 2 NMDS plots of overall gut microbiota composition stratified per regimen and time point.** Non-metric multidimensional scaling (nMDS), based on Bray–Curtis (BC) dissimilarity between samples, visualizing the overall gut microbial community composition stratified for antibiotic-treated infants and controls (**a**, **b**) and per regimen compared with controls (**d**, **e**), for the time points before and immediately after cessation of antibiotic treatment. Each data point represents the microbial community composition of one sample. The ellipses represent the standard deviation of data points belonging to each group, with the center points of the ellipses calculated using the mean of the coordinates per group. The stress of the ordination, effect sizes ($R^2$) calculated by permutational multivariate analysis of variance (PERMANOVA)-tests and corresponding adjusted $p$-values (p.adj, calculated using the Benjamini-Hochberg method) are shown in the plots. In panels **c** and **f** the results of the PERMANOVA-tests for the later time points are summarized. Ac = amoxicillin + cefotaxime (yellow), cg = co-amoxiclav + gentamicin (pink), pg = penicillin + gentamicin (purple). Data points of controls are colored teal and those of antibiotic-treated infants (not-stratified per regimen) orange. Source data are provided as a Source Data file.

controls and antibiotic-treated infants. Among others, *Bifidobacterium* spp. were less abundant in the antibiotic-treated infants compared to the controls from day 1 until day 36, also after adjusting for mode of delivery (p.adj = 0.005; Supplementary Table 4). Likewise, *Escherichia* and *Staphylococcus* spp. were less abundant in antibiotic treated infants from 1–181 and 1–229 days, respectively. Also, 28 taxa belonging to the genus *Bacteroides* were less abundant in antibiotic-treated infants compared with the controls. In contrast, antibiotic-treated infants showed, among others, higher abundances of *Klebsiella* (day 1–122) and *Enterococcus* spp. (day 1–121, p.adj = 0.02 and 0.005, respectively).

MGS of a subset of samples from infants directly following antibiotic treatment ($n = 20$; median age: 6 days) and week 1 samples from controls ($n = 12$, median age: 6 days) confirmed these significant differences in abundance of species. These data also confirmed the annotation of *B. adolescentis*, *B. breve* and *B. longum* strongly correlating with the *Bifidobacterium* OTU,

*E. coli* correlating with the *Escherichia coli* OTU, *S. epidermidis* correlating with the *Staphylococcus epidermidis* OTU, *K. pneumoniae* and *K. oxytoca* correlating with the *Klebsiella* OTU, and *E. faecium* correlating with the *Enterococcus faecium* OTU (Supplementary Fig. 4 and Supplementary Table 5). MGS also confirmed the effects of antibiotics on the overall microbial community composition, with a similar overall effect size (PERMANOVA test, $R^2$ 8.0%, $p = 0.01$) as found with the 16S-based sequencing data ($R^2$ 9.5%).

Stratified analyses for the three antibiotic regimens showed that *Bifidobacterium* was decreased in all three treatment groups for similar periods of time when compared to the controls (Supplementary Table 4). Furthermore, *Enterococcus* spp. were increased in abundance for similar periods of time in all three treatment groups. However, other taxa were more variably affected in abundance by the three different regimens. For example, *E. coli* abundance was reduced in both the amoxicillin + cefotaxime and penicillin + gentamicin regimens (day 1–142 and

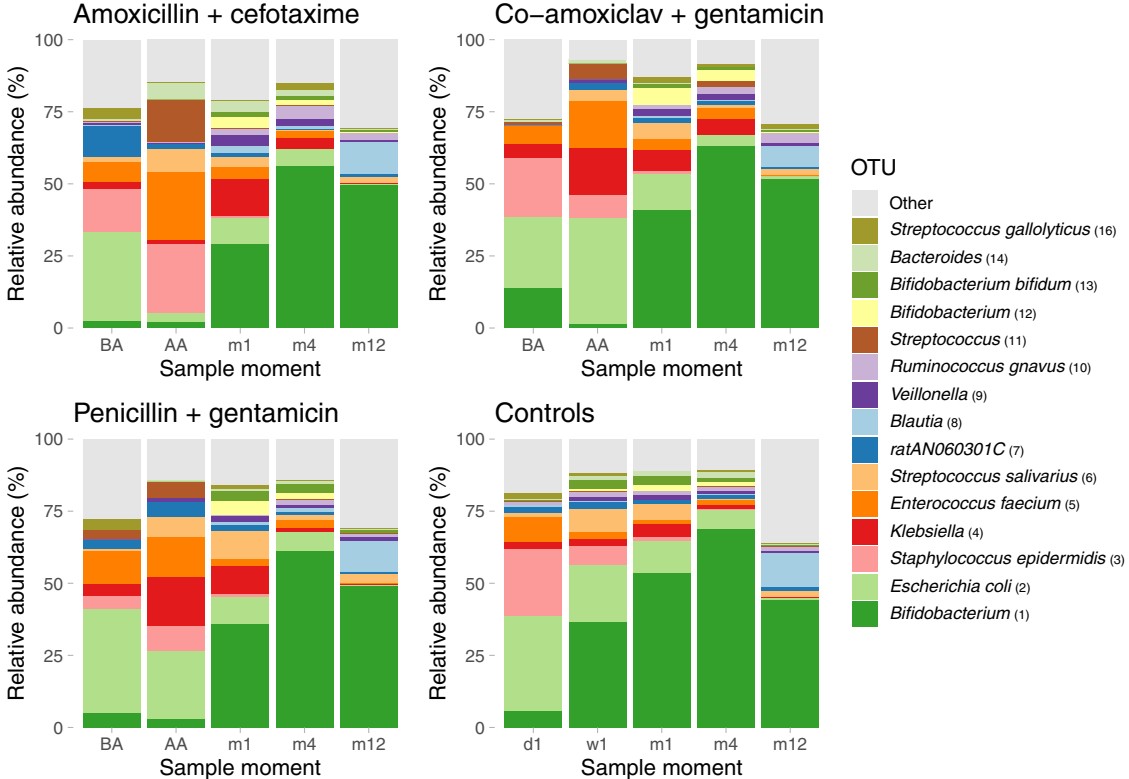

**Fig. 3 Mean relative abundance of most abundant OTUs.** Mean relative abundances of the 15 most abundant OTUs are depicted for all samples per time point, stratified by antibiotic regimen. BA = before antibiotics, AA = after antibiotics, d = day, w = week, m = month. In some cases, multiple OTUs of individual bacterial species were identified, so OTUs are referred to by their taxonomical annotations and a rank number (shown in parentheses), which is based on the abundance of each given OTU in the overall dataset. Source data are provided as a Source Data file.

day 1–177, respectively), but not in the co-amoxiclav + gentamicin regimen. *Akkermansia spp.* were generally only reduced in the amoxicillin + cefotaxime group. In contrast, *Klebsiella* abundance was higher in both the co-amoxiclav + gentamicin and penicillin + gentamicin groups, relative to the amoxicillin + cefotaxime group in early life (day 3–17 and 2–16 respectively), whereas this reversed at a later age (Supplementary Tables 6–8). Moreover, *Acinetobacter* spp. abundance was increased in the amoxicillin + cefotaxime and penicillin + gentamicin groups, but not the co-amoxiclav + gentamicin group.

Comparisons between the three different regimens, this time adjusted for antepartum maternal antibiotics, showed, among others, that the abundance of *Bifidobacterium* genera was more reduced in the amoxicillin + cefotaxime group compared to the penicillin + gentamicin group (with the exception of *Bifidobacterium animalis*, Supplementary Tables 6–8).

Though we did not find a significant difference in duration of breastfeeding between the antibiotic-treated infants and controls, we did identify an effect of breastfeeding duration on microbiota development. Though this effect was smaller than antibiotic exposure itself ($R^2$ 0.7% versus 1.4% for antibiotic exposure), we sought to test the potential modifying effect of breastfeeding on our results. We performed a post-hoc analysis to test for differentially abundant taxa between the amoxicillin + cefotaxime group and the controls, with breastfeeding added as a covariate (next to the already included mode of delivery variable), as the difference in duration of breastfeeding was the greatest between these two groups. The results can be found in Supplementary Tables 9 and 10. In summary, we found highly similar results for the analysis with or without adjusting for the additional breastfeeding covariate. Importantly, the most abundant *Bifidobacterium* OTU was still less abundant in the infants treated with

amoxicillin + cefotaxime versus controls when correcting for breastfeeding, with a similar interval and area of differential abundance identified compared to the analysis without correcting for breastfeeding.

**Effects of early-life broad-spectrum antibiotics on infant gut antimicrobial resistance gene profiles.** All 31 AMR genes included in the Fluidigm qPCR assay were detected in this study. The AMR genes least often detected were the aminoglycoside resistance gene *aph(2″)-I(de)* and the vancomycin resistance gene *vanA*, which were both present in only 6/939 samples, while the gene that was most commonly present was the multidrug efflux pump unit *acrA*, found in Gram-negative bacteria (708/939 samples)[19]. Other commonly observed genes, present in >80% of all infants, were the aminoglycoside resistance genes *aac(3′)-Ii(acde)*, *aac(6′)-Ii*, *aph(3′)-III*, and the *aadE-like* gene, the beta-lactamase encoding genes *bla_{ampC}* and *bla_{CTX-M}*, the macrolide resistance gene *ermB* and the tetracycline resistance gene *tetQ*.

The AMR gene diversity, i.e. number of observed AMR genes detected within a sample, only differed significantly between the antibiotic-treated infants and controls at 1 month of age (Wilcoxon test, median observed number of genes in antibiotic-treated infants 9, versus 7.5 in controls, $p = 0.02$). Temporal analysis showed no significant difference in the diversity of AMR genes between the antibiotic-treated group and controls over the first year of life. Also, no significant inter-regimen differences were found in cross-sectional nor temporal analyses.

On the level of AMR gene composition, or AMR gene profile, significant differences were found at multiple time points between antibiotic-treated infants and controls and also between the separate regimens compared with controls. Importantly,

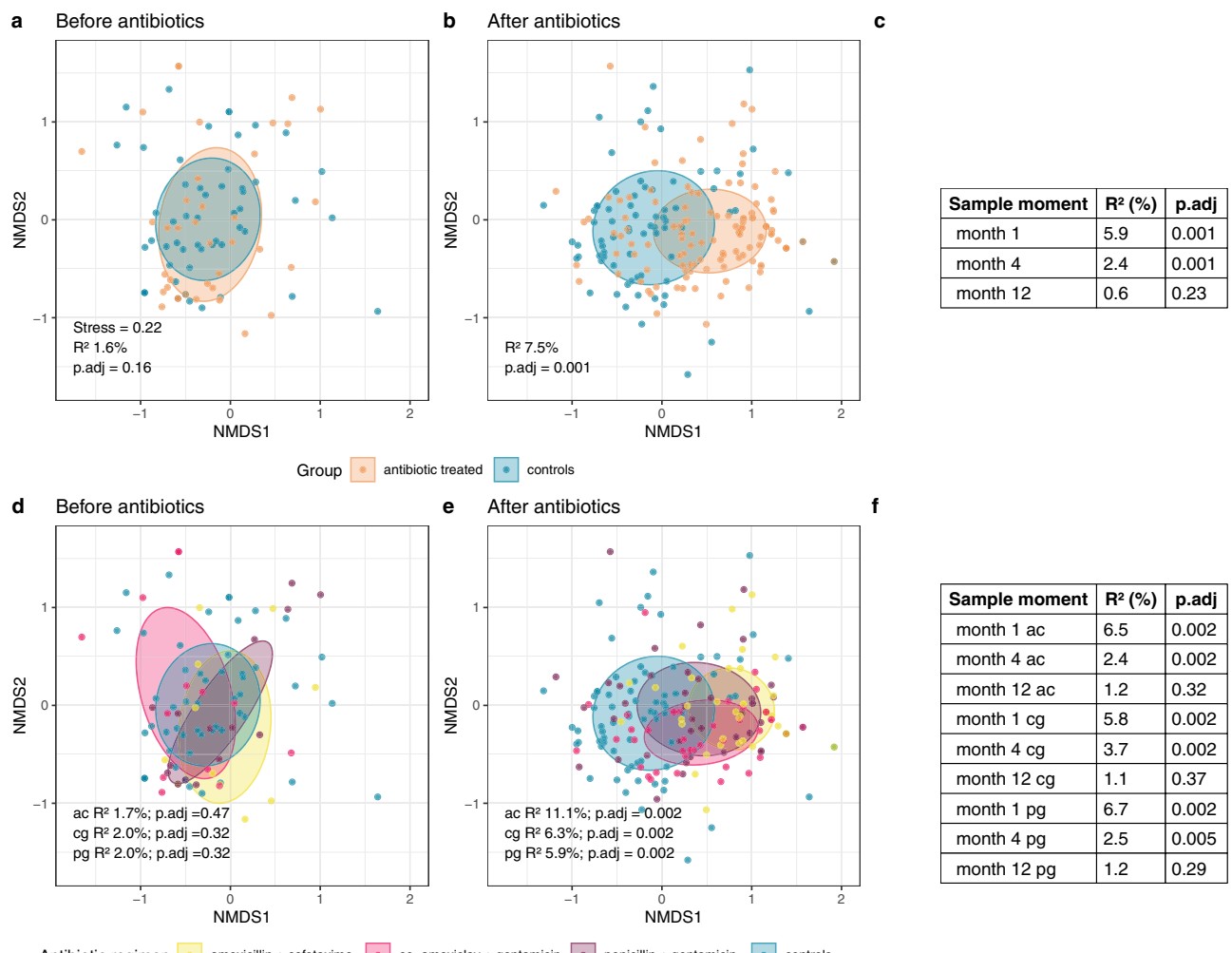

**Fig. 4 NMDS plots of AMR gene profile stratified per regimen and time point.** Non-metric multidimensional scaling (nMDS) plots, based on Jaccard index of binary (presence/absence) data, visualizing the antimicrobial resistance (AMR) gene profiles stratified for antibiotic-treated infants and controls (**a**, **b**) and per regimen compared with controls (**d**, **e**), for the time points before and immediately after cessation of antibiotic treatment. Each data point represents the composition of AMR genes of one sample. The ellipses represent the standard deviation of data points belonging to each group, with the center points of the ellipses calculated using the mean of the coordinates per group. The stress of the ordination, effect sizes ($R^2$) calculated by permutational multivariate analysis of variance (PERMANOVA)-tests and corresponding adjusted $p$-values ($p$.adj, calculated using the Benjamini–Hochberg method) are shown in the plots. In panels **c** and **f** the results of the PERMANOVA-tests for the later time points are summarized. Ac = amoxicillin + cefotaxime (yellow), cg = co-amoxiclav + gentamicin (pink), pg = penicillin + gentamicin (purple). Data points of controls are colored teal and those of antibiotic-treated infants (not-stratified per regimen) orange. Source data are provided as a Source Data file.

before the start of antibiotic treatment in the infants with sEONS, the gene profiles largely overlapped between the to-be treated neonates and controls (PERMANOVA, $R^2$ 1.6%, $p$.adj = 0.16; Fig. 4a). After antibiotic treatment, however, a major shift was observed in AMR gene profile between the treated infants and controls ($R^2$ 7.5%, $p$.adj = 0.001, Fig. 4b). The AMR gene profile slowly normalized over time (1 and 4 months; $R^2$ 5.9% and 2.4%, respectively, both $p$.adj = 0.001), and did not show a significant difference at 12 months of age ($R^2$ 0.6%, $p$.adj = 0.23, Fig. 4c). The regimen amoxicillin + cefotaxime showed the biggest effect on AMR gene profile, with an $R^2$ of 11.1% ($p$.adj = 0.002, Fig. 4e) immediately after antibiotic treatment versus $R^2$ of 6.3 and 5.9 for the co-amoxiclav + gentamicin and penicillin + gentamicin, respectively. The co-amoxiclav + genta-micin regimen had most persistent effects (at 4 months $R^2$ 3.7%, $p$.adj = 0.002, versus $R^2$ 2.5% for penicillin + gentamicin and 2.4% for amoxicillin + cefotaxime, Fig. 4f). Again, to exclude potential confounding due to antibiotic treatment received during follow-up, we performed a post-hoc analysis on the

subset of sEONS and control infants who had not received antibiotics during follow-up. Again, the effect size of antibiotics administered in the first week of life on AMR gene composition was in this subset at all time points comparable to or larger than in the primary analysis (Supplementary Table 11).

**Clinical covariates associated with antimicrobial resistance gene profiles.** Covariates that were significantly associated with AMR gene profile during follow-up are summarized in Supplementary Table 12. Gravidity and duration of ruptured membranes before birth were both associated with mode of delivery (Kruskal–Wallis test, $p = 0.02$ and Wilcoxon test $p < 0.001$, respectively). Consequently, as mode of delivery and the presence of siblings <5 years of age were the only variables that both differed between the groups at baseline and had a significant effect on AMR gene composition, these variables were corrected for in all downstream resistome analyses when comparing antibiotic-treated infants and controls. In the inter-regimen comparisons,

we adjusted for antepartum maternal antibiotics. In line with microbial community composition, we decided to perform stratified analyses into the effect of antibiotic treatment duration categorized as short (1–4 days) or long (>4 days) on AMR gene composition, and again confirmed no significant difference in AMR gene composition at any time point (Supplementary Table 13).

**Differences in antimicrobial resistance gene abundances between antibiotic-treated infants and controls.** In a temporal analysis, 16 out of 32 AMR genes were found to be significantly differentially abundant between the antibiotic-treated infants and controls (Supplementary Table 14). Ten of these were enriched in the antibiotic-treated group, and consisted of the aminoglycoside resistance genes *aac(6′)-aph(2″)*, *aac(6′)-Ib*, *aac(6′)-Ii* and *aph(3′)-III*, the beta-lactamases *bla*<sub>CMY-2</sub>, *bla*<sub>TEM</sub>, the penicillin binding protein 2A gene *mecA*, macrolide resistance genes *ermB* and *ermC* and the colistin resistance gene *mcr-1*. The time intervals in which the differences existed varied from 68 days (*aac(6′)-Ib*) to 371 days ($bla_{CMY-2}$). Six AMR genes, such as the commonly present *tetQ* gene, were more abundant in conrols[20].

When comparing each separate antibiotic regimen with the control group, we found that the penicillin + gentamicin group had, out of the tested panel, the lowest amount of enriched AMR genes (five), versus ten in both the amoxicillin + cefotaxime and co-amoxiclav + gentamicin groups (Supplementary Tables 15-17). The genes *aac(6′)-aph(2)*, $bla_{CMY-2}$, *ermB*, *ermC* and *mecA* were consistently more abundant in each of the three regimens compared to the controls. In the inter-regimen comparisons, again the penicillin + gentamicin regimen resulted in the enrichment of the lowest number of AMR genes compared to the other two regimens (two AMR genes, versus nine for the amoxicillin + cefotaxime and five for the co-amoxiclav + gentamicin group, Supplementary Tables 18-20). Unexpectedly, the amoxicillin + cefotaxime group showed an enrichment of the aminoglycoside resistance genes *aac(6′)-Ii* and *aph(3)-Ia, -Ic* compared to the penicillin + gentamicin group and of the *aadE* gene compared to the co-amoxiclav + gentamicin group.

When studying the correlation between the relative abundance of OTUs and AMR genes (Supplementary Table 21), we found that the abovementioned gene *aac(6′)-Ii* had the strongest positive correlation with *E. faecium* (Pearson's $r$ −0.50, *p*.adj < 0.001), which was highly abundant in the infants treated with amoxicillin + cefotaxime (negative coefficients here indicate a positive correlation between OTU and gene abundance, as a low Ct value indicates high abundance of an AMR gene). In this way, we could also positively correlate the *aac(6′)-aph(2)* gene, which was consistently more abundant in all three regimens compared to controls, to *E. faecium* (Pearson's $r$ −0.42, *p*.adj < 0.001). As expected, *mecA* was positively correlated with *S. epidermidis* (Pearson's $r$ −0.58, *p*.adj < 0.001). *E. coli* was most strongly positively correlated with $bla_{ampC}$ and *acrA* (Pearson's $r$ −0.69 and −0.65, both *p*.adj < 0.001), all of which were consistently more abundant in the controls.

MGS on the subset of samples resulted in 1504 AMR genes detected in this study. Using a cross-sectional differential abundance analysis, 262 genes were found to differ significantly in abundance between the antibiotic-treated and control infants, which were all more abundant in the treated infants (Supplementary Table 22). MGS confirmed the Fluidigm results of, among others, *aac(6′)-aph(2)*, *ermB*, *ermC* and *mecA* genes being enriched in the antibiotic-treated infants (zero-inflated Gaussian mixture model, log2 fold change (log2FC): 8.56, *p*.adj < 0.001; 6.10, *p*.adj = 0.001; 4.95, *p*.adj = 0.001; 6.49, *p*.adj < 0.001). Additionally, MGS identified statistically significant enrichment of AMR genes in antibiotic-treated infants not part of our Fluidigm panel, including a broad set of genes coding for resistance against rifampin, fluoroquinolones, aminocoumarins, daptomycin, elfamycins, trimethoprim, sulfonamides, fusidic acid and bacitracin (Supplementary Table 22).

## Discussion

Suspicion of EONS is a highly common indication for broad-spectrum antibiotic treatment in neonates in the first days of life[12,13]. However, in this early period of life, the neonate is also seeded with its first microbes, allowing rapid assembly of its very own unique microbial ecosystem. Therefore, broad-spectrum antibiotics in this life phase might have a significant impact on the development and ultimate composition of the infant microbiome, with potential short and long-term consequences. We studied the ecological effects of antibiotic treatment in early life, and also attempted to identify the regimen which causes least ecological harm. To this purpose, we randomized sEONS infants over three commonly used broad-spectrum antibiotic regimens.

We show major effects of broad-spectrum antibiotics on microbial diversity, community composition and AMR gene selection. Although in most neonates antibiotics were already aborted after 48 h, the impact was still measurable at 12 months of life. We also observed marked differences between the three different antibiotic regimens, suggesting that the combination of penicillin + gentamicin causes the least ecological damage.

We observe much more outspoken and prolonged effects of antibiotics than anticipated based on previous studies[21–23]. However, the current body of evidence mostly originates from older children and adults, in whom a more established, and thus stable and resilient microbiome may be present. Therefore, we hypothesize that through the early interruption of microbiome assembly and development, our treated infant population did not have a "normal" state to return to ref. [22]. Following, these young infants may have difficulty in regaining typical commensal microbiota from their environment in order to restore a natural developmental trajectory, which might explain the limited compensatory effect of breastfeeding following antibiotic-induced disruption of the microbiota. Relevant infant studies which can corroborate these findings are scarce, and usually performed in preterm infants, with short follow up[24,25].

Given the extensive and prolonged effects of the studied early-life antibiotic regimens upon the infants' microbiome, the consequences for the natural process of immune priming and maturation might also be considerable. In the last decade, microbiome-based studies have shown a clear relationship between microbiome perturbation, especially in early life, and inflammation-driven health problems[26]. For example, a reduced α-diversity of faecal microbiota has been associated with the development of allergic disease and diabetes later in life[27–30]. Given the extent of microbiota perturbations observed in our study, this warrants further research.

On the level of individual taxa, we found especially various beneficial *Bifidobacterium* spp. to be heavily affected by antibiotic treatment. Continued breastfeeding did not appear to compensate the decreased *Bifidobacterium* abundance, which might be a result of elimination of, rather than reduction in, these species as a consequence of early-life antibiotic treatment. Bifidobacteria are known to promote gut health and provide defence against pathogens[31]. These bacteria are also essential for the digestion of (human) milk oligosaccharides[32], which in the first 4–6 months of life are the sole food source for infants. Hence, potential effects on infant growth and development are imaginable when the abundance of these bacteria is decreased. Also, the extensive outgrowth of potential pathogenic bacteria

such as *Klebsiella* and *Enterococcus* spp. warrants further studies into the susceptibility to infections following early-life antibiotic treatment. In a previous study, antibiotics in early life have been associated with increased rates of diarrhea in early childhood, which we now hypothesize might be a consequence of persistent microbial dysbiosis[33].

Importantly, when comparing the antibiotic regimens, we found that all three showed significant ecological side effects. However, the amoxicillin + cefotaxime regimen showed the most outspoken ecological effects, especially on overall microbial community composition, stability of microbiota development, and AMR gene profile shifts directly following treatment. The co-amoxiclav + gentamicin regimen showed the most obvious reduction in microbial diversity and most persistent effects on AMR gene profile. Importantly, the penicillin + gentamicin regimen showed the least detrimental effects on all these parameters. In part, this can be explained by the minimal penetration into the gut lumen of aminoglycosides given intravenously[34]. While this regimen might clinically be the least popular due to the frequency of penicillin administration and the need for gentamicin serum level monitoring, we think this regimen deserves reconsideration for the treatment of sEONS on neonatal wards given the uniformly lower ecological side-effects observed, and considering the three regimens compared are equally adequate treatments for this indication[14]. Interestingly, some AMR genes were more present in controls. While this may be counter-intuitive, it is known that many commensals carry AMR genes, which we confirmed by correlating the AMR genes enriched in controls with the presence and abundance of commensal bacteria, such as the $bla_{AMPC}$ gene in *E. coli*[20,35].

There is a rapid increase in awareness regarding the side-effects of antibiotics on selection and development of AMR in pathogenic bacteria on a population level. Most hospitals currently enforce antimicrobial stewardship programs that ensure appropriate antibiotic therapy. However, these programs do not consider ecological side-effects yet, largely because the information is lacking. Presently, the main focus is to shorten the duration of broad-spectrum antibiotic treatment as much as possible. However, in our study, although a modest effect was observed for antibiotic treatment duration in association with microbiota composition, the effect was negligible compared to the initiation of antibiotics in the first place. This held true when testing the effect of antibiotic treatment duration in a categorized way, better reflecting the clinical setting (0–4 days versus >4 days). Currently, antibiotics are prescribed in 4–10% of all neonates, whereas only an estimated 1 in 1000 will develop a proven infection, likely resulting in unnecessary treatment of >90% of all treated children[12,13,36]. For this reason, more emphasis on improving the diagnostic accuracy of EONS is crucial, as the principle of when in doubt, there's no harm in treating appears far from true. Reducing the number of neonates in whom we initiate broad-spectrum antibiotic treatment is feasible, given the fact that prescription rates vary widely even between European countries with similar infection statistics[37,38]. This strongly suggests that guidelines underpinning these national differences should be compared to identify factors that could lead to more reserved prescription practices. Moreover, ongoing efforts to improve the prediction of EONS, for instance through the application of the EONS calculator, could further help to reduce unnecessary antibiotic treatment, even in countries with already low prescription rates[39]. Altogether, this would ideally result in—first and foremost—safe treatment guidelines, whilst simultaneously preserving early-life microbiome development at this critical stage of life.

Strengths of this study include analyzing samples before the start of antibiotic treatment to eliminate baseline differences as potential confounders. Importantly, albeit modest differences in overall microbiota composition and AMR gene profiles existed in our study between neonates with sEONS and controls at enrolment (before antibiotics), these were both non-significant and of limited relevance compared to the observed antibiotic effects. Furthermore, we used a randomized study design to compare the impact of three commonly used broad-spectrum regimens. Additionally, we validated our results through an independent MGS technique. The results of this study are highly applicable to other hospitals and countries, as generally common antibiotic regimens were compared[14].

A potential weakness of this study is its molecular epidemiological nature, and as such, causality between antibiotic treatment and observed ecological effects is not fully proven. However, the likelihood that our observations are a direct biological consequence of antibiotics is high, as we see no significant differences in microbiota composition or AMR gene profiles before antibiotic exposure, but marked and slowly waning effects following the exposure. Our findings are also in line with known microbial susceptibility patterns of observed bacterial species. Furthermore, because of ethical reasons, we had to include a separate control group, instead of integrating controls in our randomized design. Though the control group was enrolled from the same hospitals and using the same methodology, the slight difference in enrolment years may have caused minor unidentified potential confounding, though the microbiota composition and AMR gene profiles before antibiotic treatment look reassuringly similar. Also, due to clinical, practical and safety reasons we could not perform blinding. A potential selection bias may have occurred as a consequence of an above-average number of parents with a higher level of education having opted to participate in our study. This may also have affected the low percentage of inhouse smoking observed, and may therefore affect the generalizability of our results to some extent. Finally, we cannot rule out that parents of sEONS infants were more aware of possible side-effects of antibiotic treatment due to the participation in a trial studying precisely these effects, and therefore this may have affected their hesitance for antibiotic treatment at a later stage.

In conclusion, we found significant long-term effects of broad-spectrum antibiotic treatment for sEONS. We believe our data suggest that more emphasis should be put on reducing the number of neonates that receive broad-spectrum antibiotics for sEONS, and if needed, to preferably prescribe penicillin + gentamicin, as this regimen causes the least ecological side-effects.

## Methods

**Study design and participants**. We performed a randomized study in 147 infants who required broad-spectrum antibiotics for treatment of sEONS in their first week of life. Infants were recruited at the Spaarne Gasthuis Hoofddorp and Haarlem, Diakonessenhuis Utrecht and Tergooiziekenhuis Blaricum in the Netherlands (Zuigelingen En Bacteriële Resistentie na Antibiotica, ZEBRA trial). Inclusion criteria were indication for broad-spectrum antibiotic treatment due to sEONS in the first seven days of life, birth by vaginal delivery or SCS, gestational age ≥36 weeks, absence of prenatally established underlying morbidity, parental age of ≥18 years and the ability of parents to understand the Dutch or English language. A subset of healthy, term-born infants from the Dutch Microbiome Utrecht Infant Study (MUIS), served as controls[17]. We included 80 out of 120 infants from this birth cohort that were born vaginally or by SCS, had not received antibiotics in the first week of life and whose samples could be age-matched to those of the sEONS infants. Written informed consent was obtained from both parents. For pediatric studies, the Netherlands has a strict two-parent consent policy, and a deferred approval from the second parent is only granted in special circumstances, such as in acute illness where enrolment is required to occur in a short timeframe. After a thorough review procedure, the regional medical ethical committee granted approval to start enrolment in early January 2015. At that time, the medical ethics committee still had to reach a final decision on our request to allow enrolment of sEONS infants with a single parent's consent before randomization and collection of samples, and a deferred consent from the second parent (in case one parent was not yet in hospital). The ethical committee requested additional information to come to a weighted decision on this, and allowed the start of enrolment in the

meantime, as long as we used the strict parental consent rules. Permission for deferred consent was granted after the following committee meeting. Our trial was then registered in the Netherlands Trial Registry, leading to March 2015 being noted as date of registry. By that time six patients had been enrolled using the strict parental consent rules. Both ZEBRA and MUIS studies were approved by the Dutch National Ethics Committee (Netherlands Trial Registry NL4882 and NL3821). Participants did not receive compensation. The randomized trial conformed to the Consolidated Standards of Reporting Trials (CONSORT) guidelines (checklist uploaded during submission). In four cases a switch to a different antibiotic was made after blood/urine culture results became known and in 13 cases a switch to a different antibiotic was made in order for the infant to finish the treatment orally. These infants were also included in the intention to treat analysis as these practices reflect the clinical setting.

**Power calculation.** The study was initially powered by making use of previously published infant microbiota data, ensuring a power of 0.8 to detect at least significant differences in α- and -diversity between groups, and at least two-fold differences in abundance of the 25 most important OTUs[40]. For power calculations, we used data of OTUs with high and low variability and abundance, and varying effect sizes. Our power calculation was verified by the online (HMP-based) tool as soon as this became available[41]. We initially aimed to enrol 132 infants, 44 infants per antibiotic regimen, allowing a drop-out of 10%. Due to the accidental loss of a set of samples from 11 participants, approval was granted by the ethical committee to prolong enrolment in order to replace the lost samples, to ensure the power of the study. Eventually, 147 infants were enrolled, 49 per antibiotic regimen.

**Randomization and masking.** Infants were randomly allocated 1:1:1 to three most commonly prescribed intravenous antibiotic combinations for sEONS in the Netherlands, namely penicillin + gentamicin, co-amoxiclav + gentamicin or amoxicillin + cefotaxime. Blinding was not performed, as this was not deemed relevant in our observational study with shifts in the gut microbiome and AMR gene composition as primary study outcomes. The sequence with which participants were allocated to the groups was generated with the Research Manager software (Cloud9 Software BV) by an unaffiliated research nurse using 11 blocks of 12 (4:4:4) and one block of 15 (5:5:5) to enable a balanced randomization over the three regimens in the four hospitals. Allocation concealment was achieved using sealed, opaque envelopes which were delivered to the hospitals. Initial enrolment was performed by trained physicians who assigned the groups by selecting one of the sealed envelopes. Neither the research nurse who generated the allocation sequence nor the enrolling physicians were involved with the rest of the trial.

**Sample collection.** The duration of antibiotic treatment was not specified in the protocol, but was prescribed by the responsible physician depending on the clinical signs, reflecting the clinical setting. Rectal swabs and/or faeces were collected strictly before the start of antibiotic treatment (median sample age 1 day, interquartile range [IQR] 0–1 days) and 24–48 h after treatment cessation (median 32 h after cessation, IQR 27.6-40.1 h; median age 6 days, IQR 4–8 days) and at 1, 4 and 12 months of age (median 31 days, IQR 30–34 days; median 123 days, IQR 120–127 days; median 367 days, IQR 365–371, respectively). The sample moments before and immediately after antibiotic treatment corresponded with 1 day (median sample age 1 day, IQR 1–1 days) and 1 week of life (median 6 days, IQR 6–7 days) of the controls (Supplementary Fig. 5). Rectal swabs were collected using FaecalSwab™ kits (Copan Diagnostics, CA, USA) by trained physicians or research personnel before and 24–48 h after antibiotic treatment. Faecal samples were obtained at the same time points and additionally at 1, 4 and 12 months and stored in sterile faecal containers by a nurse during hospital stay or by the parents if the participant was already discharged at the later time points. Since we previously showed that rectal swabs can be used as a proxy for faeces when the latter is not available, we used rectal swabs where needed to ensure proper before-after treatment microbiota comparisons[42]. All materials were directly stored at −20 °C before being transferred (<2 weeks) to a −80 °C freezer until further laboratory processing. Perinatal characteristics of the participants were obtained at baseline and parents filled in online questionnaires about health characteristics at age 1, 4, 6, 8, 10 and 12 months (for the control infants a questionnaire was filled in at 9 months instead of at 8 and 10 months). Breastfeeding at time of sampling was defined as an infant receiving any breastfeeding at that time point (albeit exclusive breastfeeding or mixed). Exclusive formula feeding was defined as an infant not having received any breastfeeding at all.

**DNA isolation and sequencing.** Bacterial DNA was isolated from faecal material using the Mag Mini DNA Isolation Kit (LGC ltd, UK) according to the manufacturer's recommendations as previously described[43]. In short, we used ~100 μl of faeces or 200 μl of swab material, 300 μl of lysis buffer, 500 μl of 0,1 mm zirconium beads (BioSpec products, Bartlesville, OK, USA) and 500 μl of phenol saturated with Tris-HCl (pH 8.0; Carl Roth, GMBH, Germany) in a 96-wells plate, and performed an extra phenol/chloroform step. Mechanical disruption of samples was achieved using a Mini-BeadBeater-96 (BioSpec products, Bartlesville, OK, USA) for 2 min at 2100 oscillations per min. The extracted DNA was eluted in a final volume

of 60 μl of elution buffer (LGC Genomics, Germany). Further adaptations were applied to the samples collected at the first time point[44], being expected to be of low bacterial abundance and diversity, with the additional changes of using 150 μl of faeces (or 100 μl of material in the case of rectal swabs) and implementing an extra step with wash buffer 1. DNA blanks and a positive control consisting of a mix of up to six random faecal samples were used for quality control. The amount of bacterial DNA was determined by quantitative polymerase chain reaction (qPCR) as previously described using universal primers specifically designed to amplify the bacterial 16S rRNA genes (Forward: 5'-CGAAAGCGTGGGGAGC AAA-3'; Reverse: 5'-GTTCGTACTCCCCAGGCGG-3'; Probe: 5'-6-FAM-ATTA GATACCCTGGTAGTCCA-MGB-3') on the 7500 Fast Real-Time qPCR system (Applied Biosystems)[44].

For the sequencing of the V4 hypervariable region of the 16S rRNA gene, ~500 pg of DNA was amplified using F515/R806 primers and 30 amplification cycles[45,46]. Primers included Illumina adapters and a unique 8-nt sample index sequence key[45]. After amplification, quantification of the amount of amplified DNA per sample was executed with the dsDNA 910 Reagent Kit on the Fragment Analyzer (Advanced Analytical, IA, USA). Samples yielding insufficient DNA after amplification, defined as <0.5 ng/μl, were repeated with a higher concentration of template DNA. A mock control, positive control, DNA isolation blank and up to three PCR blanks were included in each PCR plate. Amplicons were pooled equimolarly and purified from 1.2% agarose gel using the Gel Extraction Kit (Qiagen, Hilden, Germany). The library was quantified using the Quant-iT™ PicoGreen® dsDNA Assay Kit (Thermo Fisher Scientific, MA, USA). 16S rRNA sequencing was performed on the Illumina MiSeq platform (Illumina, Eindhoven, the Netherlands).

**Bioinformatic processing.** The samples and their sequences described in this manuscript are part of a larger dataset existing of 2176 faecal/rectal swab samples and controls, and together were processed using our in-house bioinformatics pipeline[47]. In short, we applied an adaptive, window-based trimming algorithm (Sickle, version 1.33) setting the quality threshold to $q = 30$ and length threshold to 150 nucleotides[48]. Error correction was conducted with BayesHammer (SPAdes genome assembler toolkit, version 3.5.0)[49]. Each set of paired-end sequence reads was assembled using PANDAseq (version 2.10) and demultiplexed with QIIME (version 1.9.1)[50,51]. Singleton and chimeric reads (UCHIME) were removed. OTU picking was performed using VSEARCH abundance-based greedy clustering with a 97% identity threshold[52]. OTU annotation was established using the Naïve Bayesian RDP classifier (version 2.2) and the SILVA reference database[53,54]. This resulted in an OTU-table containing 18,951 taxa in total. We created an abundance-filtered dataset selecting OTUs present at a confident level of detection (0.1% relative abundance) in at least two samples, hereafter referred to as our raw OTU-table[55]. The raw OTU-table consisted of in total 730 taxa (0.49% sequences excluded with filtering). Next, we used both the prevalence and frequency methods of the decontam package with the default threshold of 0.1 to exclude possible contaminants, discarding 35 taxa, and thus retaining 695 taxa in total[56].

For the purpose of this manuscript, 989 samples from 225 infants (90.7% of obtained samples from 145 antibiotic-treated and 80 control infants) were eligible for microbiota analysis (Supplementary Fig. 6). 16S rRNA-based sequencing of the V4 hypervariable region resulted in 79,730,049 high quality reads with a minimum Good's coverage of 98.97% (median 99.99%). The raw OTU-table of the 989 samples contained 692 bacterial OTUs distributed over 18 bacterial phyla, with Actinobacteria being the most abundant phylum and Bifidobacterium the most abundant genus.

**Metagenomic shotgun sequencing and processing.** To validate our 16S rRNA sequencing and qPCR data, we performed MGS on a randomly selected subset of 32 faecal samples collected from 20 antibiotic treated infants post antibiotics and 12 controls at 1 week of age. Sample libraries were prepared using the Truseq Nano gel free library preparation kit. Using a NovaSeq instrument, 150 base paired-end sequence data was generated yielding 1500 M reads (two runs). Reads were trimmed using Cutadapt[57] (version cutadapt-1.9.dev2) maintaining a quality threshold of 30 and a minimum read length of 35 base pairs. Per-sample and per-run SAM files were generated using Bowtie2 and the MetaPhlAn2 and MEGARes databases[58–60], while adhering to recommended parameters, including suppressing unaligned reads. Each SAM file was assigned a read group and SAM files from different runs were merged sample-wise using Picard[61]. Merged SAM files for each sample were sorted and indexed using SAMtools[62]. Raw reads and counts of reads mapped to each AMR gene were generated using SAMtools idxstats. To calculate counts per million in order to normalize read counts for library size, counts were divided by a normalization factor of the library size divided by 1,000,000.

**Identification of AMR genes by qPCR and processing.** The primer set used in the qPCR assays covered 31 AMR genes, including genes encoding extended-spectrum β-lactamases, carbapenemases and proteins involved in vancomycin resistance (Supplementary Table 23). The selection of AMR genes was based on two previous studies, aimed to detect the most common AMR genes in the gut microbiota of healthy individuals and clinically relevant AMR genes[10,63]. Of the 81

genes covered in these studies, we selected a total of 31 that were detected in ≥1 sample in either study and were deemed clinically relevant for our pediatric population and the antibiotic regimens being compared.

For the identification of AMR genes through qPCR we used the 96.96 BioMark™ Dynamic Array for Real-Time PCR (Fluidigm Corporation, San Francisco, CA). The concentration of each sample was measured using a specific 16S qPCR and all samples were normalized to 0.1 ng per ul. Any sample below this threshold was included in an undiluted state. Samples were pre-amplified with all primers for a total of 14 cycles, according to the manufacturer's instructions, excluding the 16S rRNA primer, due to the high abundance of 16S rRNA present in the samples, compared to the low abundance of the AMR targets. Primers were used at a concentration of 500 nM. The multiplex qPCR was run using the manufacturer's recommended settings, with the exception of the melting temperature being held at 56 °C and the total number of cycles set to 35, which were adjusted according to the melting temperatures of the primers used, previous optimization and our own internal validation[10,63]. All samples were run in triplicate. On all chips we included the 16S rRNA gene as the reference gene, two positive controls (a sample pool existing of all samples from chip plates 1–5, and a hospital sewage sample) and Non-Template Controls (NTCs). We standardized the within-individual variation across time points by including all time points from a given individual on a single chip. Individuals were then randomized across chips. Ct values were extracted using the BioMark Real-Time PCR analysis software. NTCs and positive controls were checked within and between chips for errors and consistency. The detection limit on the Biomark system was set to a Ct value of 20 for AMR targets and 34 (because of lack of pre-amplification) for the 16S RNA reference gene. Sample-primer combinations that were failed by the software were checked manually and either passed or failed according to the suitability of the melting and amplification curves. Melting curve analysis was performed for all reactions within the Biomark Real-Time PCR analysis software to ensure the validity of the measured biological signals. Melting curves of the reactions were compared to those of the positive controls, and reactions showing unreliable S-curves were discarded. A sample was only included when a minimum of two of its triplicate reactions resulted in a Ct value below the detection limit, and for each included sample we calculated an average Ct value. Delta Ct values were calculated using the formula CtAMR—Ct16S rRNA. To account for the preamplification step of the AMR primers and to avoid negative delta Ct values, we added 20 to this value. Depending on the downstream statistical analysis, we transformed the dataset to include binary information or continuous Ct values with an arbitrary value of 32 appointed to negative sample-primer reactions.

Of the 989 samples that were included in the microbiota analyses, 939 samples passed the Fluidigm BioMark™ (Dynamic Array for Real-Time PCR) quality check and were included in the resistome analyses (Supplementary Fig. 6).

**Statistical analysis**. All analyses were performed in R version 3.4.3 within RStudio version 1.1.383 and figures were made using package ggplot2 and ggpubr[64–67]. A statistical analysis scheme showing the flow in and order of analyses to address the primary research questions can be found in Supplementary Fig. 7.

For the microbiome analyses, the numbers of samples analyzed of the control group at time point 1 day, 1 week, 1 month, 4 months and 12 months, were: 66, 80, 80, 78 and 74, respectively. For the amoxicillin + cefotaxime treated infants the numbers of samples analyzed at time point before antibiotics, after antibiotics, 1 month, 4 months and 12 months, were: 17, 46, 46, 45 and 44, respectively; for the co- amoxiclav + gentamicin treated infants this was: 20, 45, 47, 46 and 44, respectively; and for the penicillin + gentamicin treated infants this was: 23, 48, 47, 46 and 45, respectively. For the resistome analyses, the numbers of samples analyzed of the control group at abovementioned time points, were: 55, 76, 78, 77 and 72, respectively; for the amoxicillin + cefotaxime treated infants this was: 11, 40, 46, 45 and 43, respectively; for the co-amoxiclav + gentamicin treated infants this was: 13, 41, 46, 43 and 42, respectively; and for the penicillin + gentamicin treated infants this was: 15, 46, 46, 45 and 45, respectively.

For simple, independent comparisons, we considered p-values < 0.05 to be significant. However, for all analyses regarding multiple comparisons, we applied the Benjamini–Hochberg method to correct for multiple testing[68]. For comparisons of group differences, a one-way analysis of variance test, Wilcoxon rank-sum test, Kruskal–Wallis test, or chi-square test was used, where appropriate.

Group differences in Shannon α-diversity and observed number of genes were calculated using Wilcoxon tests and linear mixed-effect models with participant set as random effect while correcting for age.

The overall gut microbial community composition of faecal samples and rectal swabs was visualized using non-metric multidimensional scaling plots (nMDS; vegan package[69]). Ordinations were based on the BC dissimilarity matrix of relative abundance data with parameter trymax 10,000.

Associations between clinical outcome and microbiota composition were analyzed with the adonis2 function (vegan package[69]), based on PERMANOVA-tests per time point and across all time points using 1999 permutations using relative abundance data. To test for multivariate spread, we used the function betadisper (vegan package[69]). The dispersion of the data of the compared groups was significantly heterogenous immediately after antibiotic treatment onwards for the antibiotic-treated versus non-treated comparisons ($p < 0.001$, $p < 0.001$,

$p = 0.02$ and $p = 0.02$, respectively), while this was only the case immediately after antibiotics and at 1 month of age for the comparisons between the separate regimens and controls. In the inter-regimen comparisons, the dispersion of the data was homogenous at all time points. In the multivariable, temporal analyses all variables were included that showed a significant association with microbiota composition when tested individually, and age and subject were added to control for repeated measures.

The stability of the gut microbiota composition in the first year of life was visualized by measuring the BC dissimilarities between consecutive samples within each participant over time. To test for group differences, the Wilcoxon test was used.

Individual bacterial taxa and their succession patterns, and potential differences thereof between groups, were studied at the lowest taxonomic annotated level (OTU). Differential abundance testing was executed with smoothing spline analysis of variance (SS-ANOVA, fitTimeSeries function, metagenomeSeq package[70,71]) allowing not only to detect biomarker OTUs related to antibiotic treatment but also to identify the specific time intervals in which significant differences existed. For this analysis, the raw OTU-table was filtered and OTUs with >10 reads in ≥50 samples were included, resulting in 512 OTUs (of the 692)[47]. This filtering step was performed separately for each subset of data used for the different comparisons. The SS-ANOVA analysis was adjusted for covariates that both had (1) an effect on overall gut microbiota composition and (2) were unevenly distributed between the antibiotic-treated infants versus controls, namely mode of delivery, and the inter-regimen comparisons were adjusted for antepartum maternal antibiotics. Although gravidity of mothers, gestational age, duration of ruptured membranes, hospital stay duration after birth, presence of siblings <5 years of age and inhouse smoking were all associated with antibiotic-treated infants, we did not find an association between these clinical variables and gut microbiota composition, therefore these six variables were excluded as covariates from downstream fitTimeSeries analyses. While day care attendance and breastfeeding at the time of sampling were associated with gut microbiota composition, these variables did not differ at baseline and therefore were also excluded as covariates from the downstream analyses.

Results from 16S rRNA sequencing at 1 week of life were validated by untargeted MGS (subset of 32 samples). The diversity of AMR genes was defined as the observed number of different AMR genes present in a sample. Comparisons in AMR diversity were calculated using the Wilcoxon test. The overall AMR gene composition, or profiles, of faecal samples and rectal swabs were visualized as described above, with the exception that the ordination was based on the Jaccard index of the binary data. Associations between clinical outcome and AMR gene profiles were analyzed similarly to the 16S data, but again using binary data and the Jaccard index. Dispersion of data was only significantly heterogeneous in one case, namely at the time point immediately after antibiotic treatment in the comparison of the amoxicillin + cefotaxime regimen versus controls.

Differential abundance testing of the AMR genes was also executed with SS-ANOVA (fitTimeSeries function, metagenomeSeq package[70,71]), though no filtering was performed to allow for testing of all 31 AMR genes and no normalization was applied in the function. The SS-ANOVA analysis was adjusted for covariates that both had (1) an effect on overall AMR gene composition and (2) were unevenly distributed between the groups compared, so for siblings <5 years of age and mode of delivery in the antibiotic-treated infants versus controls and for antepartum maternal antibiotics in the inter-regimen comparisons.

We evaluated the correlation between 16S OTUs and AMR genes by calculating the Pearson correlation coefficient for all individual taxa based on their relative abundance. Positive coefficients indicated a negative correlation between OTU and gene abundance, as a high Ct value indicates low abundance of an AMR gene. The fitZig function of the metagenomeSeq package[71] was used to assess the group differences in AMR genes as identified by MGS in a subset of 32 samples, after removing rare features present in <10 samples, resulting in 343 out of 1504 AMR genes included in this analysis.

**Reporting summary**. Further information on research design is available in the Nature Research Reporting Summary linked to this article.

## Data availability

Sequence data that support the findings of this study have been deposited in the NCBI Sequence Read Archive (SRA) database with BioProject IDs PRJNA481243, PRJNA524461, PRJNA555020, PRJNA612836, PRJNA613054 and PRJNA613032. Source data underlying Figs. 1–4, Supplementary Figs. 2–4 and Supplementary Tables 1–3, 5 and 11–13 are provided with this paper. Our study protocol was provided on submission.

## Code availability

No custom code was used for the creation of this manuscript.

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

## Acknowledgements

The authors are grateful for the participation of all the children and their families. We wish to acknowledge all the members of the research team of the Spaarne Gasthuis Academy and the Neonatology Departments of the Spaarne Gasthuis Hoofddorp and Haarlem, Diakonessenhuis Utrecht and Tergooiziekenhuis Blaricum for the help in participant recruitment, and the laboratory staff, with special thanks to Raiza Hasrat of the University Medical Center Utrecht. We would also like to acknowledge Edinburgh Genomics for executing the MGS sequencing and Katherine Emelianova for the bioinformatic processing of the MGS sequences. This research was funded by the Netherlands Organisation for Health Research and Development (ZonMw Priority Medicines Antimicrobiële Resistentie project number 205300001, MR/MAvH), Chief Scientist Office (SCAF/16/03, DB) and the Spaarne Gasthuis (MR/MAvH). The funder of the study had no role in study design, data collection, data analysis, data interpretation, or writing of the report. The corresponding author had full access to all the data in the study and had final responsibility for the decision to submit for publication.

## Author contributions

M.A.v.H., R.J.L.W., E.A.M.S. and D.B. conceived and designed the experiments. M.R., M.A.v.H., W.J.d.W., I.S. and F.B.P. included the participants. R.L.W., M.L.J.N.C. and K.A. were responsible for the execution and quality control of the laboratory work. M.R. and D.B. analyzed the data. M.R., M.A.v.H., W.v.S., E.A.M.S. and D.B. wrote the paper. All authors significantly contributed to interpreting the results, critically revised the manuscript for important intellectual content and approved the final manuscript.

## Competing interests

D.B. declares to have received unrestricted fees paid to the institution for advisory work for Friesland Campina as well as research support from Nutricia. None of the fees or grants listed here was received for the research described in this paper. No other authors report financial disclosures. The authors declare no competing interests.
