## [Peer Review File · Nature Communications]

REVIEWER COMMENTS

Reviewer #2 (Remarks to the Author):

The reviewer thanks the authors for addressing all comments raised by the reviewer. I think most have been answered satisfactory.

Although the reviewer acknowledges that the authors prefer to describe the data and analyses in a similar way to previous papers from their group, some factors such as duration of (exclusive) breastfeeding has always been reported as a major factor in microbiota development and therefore should be included to compare with other studies. The fact that the authors do not find a significant impact of duration of (exclusive) breastfeeding suggests that breastfeeding alone is not able to compensate the impact of AB in the first week on microbiota development?

The reviewer is a bit puzzled about the fact that whereas in the first version of the manuscript significant differences in % later AB were found in one of the groups and when they went back to review their data 4 more cases emerged in this group resulting in non-significant differences. How did the authors monitor their data from the parent reports?

Finally if the reviewer has read the revised version correctly about half of the infants in the groups received AB for 1-4 days and the other half 5-7 days? It is remarkable that in 2 groups the median is < 3 and in the third group it is 4.5 suggesting that in the first 2 groups most infants received 2-3 days AB and in the third group most received >4 days or is the distribution skewed and did those who received longer AB mostly 7 days. My question is that in several studies prolonged i.e. >4 days AB is associated with more adverse health outcomes and microbiota outcomes which were not found in this study. I think this remarkable finding and the possible reasons for not finding this effect in this study whereas it is found in most other studies should be discussed in more detail in the discussion.

Reviewer #3 (Remarks to the Author):

I have gone through the revised and and resubmitted version of the manuscript by Ms Reyman and colleagues. The quality of the manuscript has improved. Still, there is one important crucial issue that should be more emphasized in the manuscript.

Most of the significant differences are found between the RCT cohort (infants receiving antibiotics) and the control cohort (infants not receiving antibiotics). I understand that it was difficult to include non-antibiotic infants in the RCT due to ethical reasons. However, it still means that there is a high risk of systematic bias. We know from other studies that the gut microbiome differs between different cohorts. In the present study the two cohorts included infants from the same hospitals but it seems, however, that the recruitment was separated by time, and the gut microbiome composition is known to fluctuate over time.

I don't think that this disqualifies this manuscript from being published, but I think that this issue should be better covered in the text.

Suggestions:

- Add the years when the control cohort was including infants in the result section.
- Add the years when the RCT and the control cohort, respectively, was including infants in the abstract
- Discuss the risk of systematic bias as described above in the discussion section.

Reviewer #4 (Remarks to the Author):

In general, they addressed all the issues raised by Reviewer 1. The only part that I still believe needs some attention is the following point regarding breastfeeding.

I totally agree with reviewer 1 regarding the inclusion of breastfeeding in the analysis, especially since bifidobacterium was found to be differential. I think the answer by the authors is good, but I would like to see that answer in the revised paper (for example in the supplemental information). Currently, it's only part of the response to the reviewers.

In addition, the authors claim that they only examined features that differentiated across the groups, but it seems like breastfeeding was indeed different, for example, at 4 months of age, so I'm not 100% sure why it was not included.

In addition, it took me a really long time to understand the type of data the paper is based on. I searched the abstract, end of introduction, and only in the methods section I realized that the vast majority of analysis is based on PCR of known AMR. Is that so? In any case, this information should be made clearer MUCH earlier in the paper. As a reader, I assumed this is metagenomic sequencing, but when I got to the region discussing the few samples that underwent further investigation with mgx sequencing, I realized something was missing.

It might be just language, but I found this statement to be confusing. I imagine/hope that all data were deposited to NCBI, and not only the subset that supported these findings, right? :)

“Sequence data that support the findings of this study have been deposited in the NCBI Sequence Read Archive (SRA) database with BioProject IDs PRJNA481243, PRJNA524461 and PRJNA555020.”

Response to the Reviewers' comments

Reviewer #2

The reviewer thanks the authors for addressing all comments raised by the reviewer. I think most have been answered satisfactory.

We thank the Reviewer for their positive response.

Specific comments

Although the reviewer acknowledges that the authors prefer to describe the data and analyses in a similar way to previous papers from their group, some factors such as duration of (exclusive) breastfeeding has always been reported as a major factor in microbiota development and therefore should be included to compare with other studies. The fact that the authors do not find a significant impact of duration of (exclusive) breastfeeding suggests that breastfeeding alone is not able to compensate the impact of AB in the first week on microbiota development?

We agree with the Reviewer that the duration of breastfeeding should be included in the analyses, even if it is indeed to underline the relative limited impact on compensating for antibiotic treatment.

First, we describe the effect of breastfeeding in our overarching multivariate analysis using the overall (line 138) and antibiotic treated cohort (line 152), showing breastfeeding at time of sampling has an effect size of 0.7% and 0.8%, respectively.

Moreover, we described in our previous rebuttal the additional fitTimeSeries analysis performed on the amoxicillin + cefotaxime group versus the control infants with breastfeeding as a covariate next to the already included mode of delivery variable, as the difference in duration of breastfeeding was the greatest between these two groups. As the fitTimeSeries analysis is longitudinal in nature, the included variable "breastfeeding at sampling moment" translates into the duration of breastfeeding given up to that time point. We conclude that most differentially abundant OTUs, including *Bifidobacterium*, *Klebsiella* and *Enterococcus* remain similarly different regarding interval and area after correcting for breastfeeding, suggesting that breastfeeding alone is not able to compensate for the impact of antibiotics in the first week of life on microbiota development. This is in line with the finding that breastfeeding alone cannot compensate for the difference in *Bifidobacterium* found between infants born by caesarean section and vaginal birth.¹ In answer to Reviewer #4, we have now incorporated the results into the manuscript at lines 207-219 and have added the tables R4 and R5 to the Supplementary Information as eTables 8 and 9. We have now added this important information to the discussion at lines 330-333 and 344-346.

Table R4. FitTimeSeries results of all significantly differentially abundant taxa between the amoxicillin + cefotaxime group and controls without adjusting for breastfeeding

OTU	Int.no	Interval start	Interval end	Area	p.adj
Bifidobacterium_1	1	1	43	- 322.223	0.00632 7
Escherichia_coli_2	1	1	142	- 383.918	0.00632 7
Staphylococcus_epidermidis_3	1	12	175	- 190.848	0.02488 3
Klebsiella_4	1	16	127	412.066 7	0.00897 3
Enterococcus_faecium_5	1	1	117	330.486 3	0.00632 7
Streptococcus_salivarius_subsp_thermophilus_6	1	1	27	- 96.3985	0.00632 7
ratAN060301C_7	1	1	343	- 797.615	0.01677
Veillonella_9	1	119	127	19.9210 5	0.00632 7
Veillonella_9	2	4	17	- 80.0658	0.00632 7
Ruminococcus_gnavus_CC55_001C_10	1	1	64	- 108.761	0.01121 6
Streptococcus_11	1	1	384	676.769 6	0.01316
Bifidobacterium_12	1	1	18	- 47.0536	0.00632 7
Bifidobacterium_bifidum_NCIMB_41171_13	1	1	213	- 630.168	0.00632 7
Clostridium_sensu_stricto_1_15	1	23	127	397.374 8	0.01121 6
Clostridium_sensu_stricto_1_15	2	1	10	- 22.6993	0.00632 7
Streptococcus_pyogenes_17	1	1	168	- 93.9667	0.00897 3
Fusicatenibacter_saccharivorans_20	1	78	384	727.045 1	0.01468 8
Clostridium_sensu_stricto_1_23	1	26	123	250.394 1	0.00632 7
Clostridium_sensu_stricto_1_23	2	364	369	7.9566	0.02785 9
Clostridium_sensu_stricto_1_23	3	1	7	- 11.5374	0.00632 7
Collinsella_24	1	1	334	- 1214.79	0.00897 3
Clostridium_butyricum_26	1	5	241	626.497 3	0.01677
Veillonella_29	1	26	289	- 382.257	0.04140 2
Bacteroides_31	1	1	384	- 820.812	0.00897 3
Peptostreptococcaceae_32	1	51	135	169.982 2	0.01592

Bacteroides_33	1	1	384	-	0.01316
				955.142	
Clostridium_butyricum_35	1	37	146	215.841	0.01121
				8	6
Clostridium_butyricum_35	2	1	5	-	0.02968
				4.74036	5
Bifidobacterium_animalis_36	1	1	24	-	0.01121
				47.4711	6
Haemophilus_38	1	43	75	78.7895	0.00632
				1	7
Haemophilus_38	2	1	13	-	0.00632
				26.2015	7
Clostridium_paraputrificum_41	1	51	369	781.940	0.01677
				6	
Corynebacterium_47	1	1	170	110.201	0.01677
Bacteroides_48	1	46	315	-	0.01121
				129.396	6
Lactobacillus_51	1	26	289	-	0.03850
				124.669	1
Streptococcus_anginosus_subsp_whileyi_55	1	1	79	-	0.00897
				98.3452	3
Bacteroides_58	1	81	384	-	0.02402
				257.989	
Bifidobacterium_63	1	1	361	-	0.00632
				626.805	7
Akkermansia_64	1	34	384	-	0.02488
				201.447	3
Bacteroides_70	1	1	335	-	0.01121
				436.998	6
Actinomyces_73	1	43	197	-	0.03850
				124.623	1
Erysipelotrichaceae_74	1	80	384	-	0.03265
				210.526	9
Veillonella_76	1	1	51	-	0.01592
				73.1769	
Corynebacterium_striatum_77	1	61	384	218.309	0.00632
				6	7
Lactococcus_lactis_78	1	1	167	122.316	0.01468
				8	8
Lachnospiraceae_86	1	203	384	316.773	0.02847
				6	2
Veillonella_87	1	1	35	-	0.01121
				32.5948	6
Stenotrophomonas_maltophilia_91	1	1	185	63.9359	0.01468
				1	8
Tepidimonas_95	1	1	15	33.0848	0.00632
				4	7
Peptostreptococcus_98	1	1	166	-	0.02075
				143.485	5
Lachnospiraceae_103	1	120	384	236.788	0.02488
				4	3
Clostridium_sensu_stricto_1_110	1	118	136	8.87793	0.02488
				7	3
Veillonella_115	1	1	13	-	0.02847

				11.0155	2
Varibaculum_117	1	64	214	- 138.246	0.04428 9
Enhydrobacter_119	1	1	33	32.6213 8	0.00632 7
Gemella_120	1	113	124	11.5253 9	0.01592
Gemella_120	2	1	32	- 74.6978	0.00632 7
Bilophila_wadsworthia_3_1_6_123	1	1	384	- 629.623	0.00897 3
Clostridium_sensu_stricto_1_124	1	1	171	122.352 7	0.02284 8
Comamonadaceae_130	1	1	27	27.8617 8	0.00632 7
Sutterella_134	1	1	381	- 305.212	0.01592
Atopobium_138	1	97	206	- 22.3409	0.01677
Acinetobacter_143	1	1	27	28.2522 3	0.00632 7
Bifidobacterium_146	1	1	148	- 147.088	0.01468 8
Akkermansia_153	1	59	308	- 99.4841	0.04084 2
Escherichia_Shigella_154	1	1	29	- 77.5914	0.00632 7
Burkholderia_156	1	1	27	19.0248 3	0.00632 7
Acinetobacter_157	1	1	178	107.878 8	0.00897 3
Collinsella_tanakaei_163	1	32	132	- 35.9938	0.04084 2
Finegoldia_164	1	1	229	189.322	0.00632 7
Dorea_170	1	369	384	49.9901 5	0.00632 7
Dorea_170	2	268	363	- 832.234	0.00632 7
Eggerthella_178	1	320	384	112.596 5	0.01468 8
Eggerthella_178	2	1	160	- 185.537	0.01316
Ruminococcus_sp_14531_180	1	1	255	100.338 6	0.03265 9
Enterococcus_186	1	1	70	56.5941	0.03727 9
Lactococcus_189	1	1	25	31.8205 4	0.00632 7
Lactobacillus_delbrueckii_subsp_bulgaricus_190	1	155	384	- 153.944	0.04428 9
Streptococcus_192	1	49	62	21.9857 8	0.01468 8
Streptococcus_192	2	1	18	-	0.00632

				35.0849	7
Bacteroides_194	1	139	384	- 193.755	0.02488 3
Akkermansia_196	1	71	350	- 107.308	0.03727 9
Klebsiella_200	1	15	132	287.952 8	0.01121 6
Veillonella_sp_DNF00869_201	1	106	120	9.47411 6	0.00632 7
Veillonella_sp_DNF00869_201	2	1	22	- 21.6387	0.00897 3
Bifidobacterium_animalis_202	1	1	51	- 28.2932	0.04354 5
Parabacteroides_distasonis_207	1	1	292	- 146.744	0.00897 3
Bacteroides_210	1	1	384	- 435.607	0.00897 3
Alistipes_215	1	31	351	- 208.304	0.01592
Bifidobacterium_228	1	1	245	- 190.287	0.02488 3
Bifidobacterium_233	1	1	189	- 123.484	0.01121 6
Bacteroides_236	1	108	384	- 205.636	0.00897 3
Collinsella_237	1	1	320	-792	0.01468 8
Actinobaculum_schaalii_FB123_CNA_2_239	1	23	143	- 114.071	0.00632 7
Bacteroides_247	1	1	312	- 227.701	0.00897 3
Bacteroides_249	1	277	384	- 69.7358	0.02402
Bifidobacterium_251	1	1	239	- 219.754	0.02075 5
Alistipes_255	1	142	384	- 135.448	0.04715
ratAN060301C_268	1	1	364	- 319.661	0.00632 7
Leuconostoc_275	1	1	26	21.0776 9	0.00632 7
Barnesiella_276	1	97	384	- 133.937	0.02847 2
Leuconostoc_mesenteroides_282	1	1	26	23.9029 4	0.00632 7
Ruminococcaceae_287	1	187	384	- 124.146	0.02785 9
Bacteroides_fragilis_CL03T00C08_291	1	13	384	- 280.553	0.01592
Actinomyces_292	1	149	271	49.6790 5	0.04140 2
Veillonella_300	1	96	129	25.6936 3	0.02847 2
Pasteurella_pneumotropica_301	1	1	14	-	0.01468

				10.2457	8
Bifidobacteriaceae_320	1	94	175	-93.106	0.01677
Escherichia_Shigella_331	1	1	9	- 7.27705	0.00632 7
Gardnerella_332	1	1	384	- 306.135	0.01592
Bifidobacterium_333	1	1	26	- 41.4159	0.00632 7
Dermabacter_335	1	1	163	- 59.2534	0.04084 2
Bifidobacterium_336	1	45	216	- 54.2568	0.02847 2
Streptococcus_338	1	1	25	10.4807 9	0.02402
Bifidobacteriaceae_344	1	1	10	- 7.41766	0.00897 3
Bifidobacteriaceae_344	2	120	134	- 12.1892	0.01592
Clostridium_difficile_630_345	1	1	11	- 3.72434	0.01316
Bifidobacterium_347	1	40	384	- 226.351	0.01677
Clostridium_sensu_stricto_1_355	1	28	129	77.5435 9	0.00897 3
Ruminococcus_gnavus_CC55_001C_364	1	1	4	- 1.75659	0.03996 7
Dialister_365	1	77	384	- 103.768	0.00897 3
Bifidobacterium_368	1	2	211	-161.21	0.00632 7
Lactobacillus_371	1	1	140	- 42.2732	0.01677
Streptococcus_galloyticus_subsp_macedonicus_386	1	1	173	- 54.0763	0.01316
Bacteroides_400	1	96	384	- 163.619	0.01121 6
Actinomyces_sp_oral_clone_DR002_406	1	18	117	116.169	0.02847 2
Corynebacterium_propinquum_408	1	1	73	-50.679	0.04277 1
Lachnospiraceae_419	1	204	384	162.777 9	0.04428 9
Staphylococcus_427	1	1	70	- 32.7776	0.02785 9
Corynebacterium_freneyi_429	1	1	143	34.6955 5	0.02968 5
Methylobacterium_radiotolerans_437	1	1	217	- 62.3588	0.04140 2
Coprobacter_443	1	49	210	- 41.0252	0.02673 2
Bacteroides_445	1	1	384	- 242.879	0.01121 6
Staphylococcaceae_452	1	1	64	35.0688 8	0.02968 5

Streptococcus_gallolyticus_subsp_macedonicus_464	1	1	20	-13.4726	0.01316
Escherichia_Shigella_469	1	1	168	-59.9257	0.041402
Bacilli_471	1	1	69	-46.4507	0.006327
Bifidobacterium_breve_473	1	1	65	-47.0128	0.006327
Bacteroides_482	1	37	111	-19.3429	0.01316
Bacteroides_482	2	165	384	-107.393	0.014688
Plesiomonas_485	1	39	200	91.10655	0.01316
Bacteroides_493	1	38	247	-80.1326	0.02402
Veillonella_502	1	1	6	-2.75991	0.01316
Corynebacteriaceae_504	1	1	13	7.0087	0.008973
Corynebacteriaceae_504	2	39	122	-64.5451	0.01677
Staphylococcus_507	1	1	19	-12.4752	0.043545
Streptococcus_521	1	1	178	-59.45	0.02402
Bacteroides_535	1	89	384	-85.344	0.018981
Staphylococcus_565	1	1	23	-23.2521	0.006327
Ruminococcus_sp_CE2_596	1	107	384	-132.235	0.020755
Bacteroides_603	1	125	384	-92.2238	0.01592
Bacteroides_645	1	1	118	28.56168	0.037279
Campylobacter_646	1	1	131	31.06805	0.043545
Bifidobacterium_animalis_647	1	9	264	52.5902	0.01677
Bacteroides_657	1	1	314	-101.568	0.006327
Bacteroides_658	1	64	384	-105.136	0.032659
Clostridium_sensu_stricto_1_673	1	42	242	49.04653	0.027859

Differential abundance testing by smoothing spline analysis of variance (SS-ANOVA) was executed to test in which specific intervals significant differences in OTUs existed between the groups, adjusted for mode of delivery. The OTUs are arranged in descending order based on their relative abundance in this dataset. A positive Area value indicates that the abundance of a specific OTU is higher in the amoxicillin + cefotaxime regimen, while a negative area value indicates that the abundance of that OTU is higher in the controls. To correct for multiple testing, the Benjamini-Hochberg method was applied, and the adjusted p-values (p.adj) are shown. Int.no = interval number. In some cases, multiple OTUs of individual bacterial species were identified, so OTUs are referred to by

their taxonomical annotations and a rank number (shown in parentheses), which is based on the abundance of each given OTU in the overall dataset.

Table R5. FitTimeSeries results of all significantly differentially abundant taxa between the amoxicillin + cefotaxime group and controls while adjusting for breastfeeding

OTU	Int.no	Interval start	Interval end	Area	p.adj
Bifidobacterium_1	1	1	43	-324.479	0.005386
Escherichia_coli_2	1	1	48	-158.265	0.005386
Klebsiella_4	1	16	125	389.7467	0.010182
Enterococcus_faecium_5	1	1	94	251.3869	0.005386
Streptococcus_salivarius_subsp_thermophilus_6	1	1	28	-101.612	0.005386
ratAN060301C_7	1	1	329	-730.462	0.02082
Veillonella_9	1	120	126	14.81322	0.005386
Veillonella_9	2	3	17	-79.5033	0.005386
Ruminococcus_gnavus_CC55_001C_10	1	1	61	-103.917	0.010182
Streptococcus_11	1	1	384	727.6015	0.018018
Bifidobacterium_12	1	1	18	-47.5474	0.005386
Bifidobacterium_bifidum_NCIMB_41171_13	1	1	207	-600.948	0.008123
Clostridium_sensu_stricto_1_15	1	23	127	388.411	0.008123
Clostridium_sensu_stricto_1_15	2	1	10	-22.6441	0.011798
Streptococcus_pyogenes_17	1	1	168	-94.5789	0.008123
Fusicatenibacter_saccharivorans_20	1	84	384	698.8715	0.028154
Clostridium_sensu_stricto_1_23	1	29	81	127.6938	0.005386
Clostridium_sensu_stricto_1_23	2	1	8	-13.0972	0.008123
Collinsella_24	1	1	342	-1325.47	0.008123
Clostridium_butyricum_26	1	4	241	590.5183	0.010182
Akkermansia_28	1	101	384	-386.684	0.046267
Bacteroides_31	1	1	384	-829.847	0.011798
Bacteroides_33	1	1	384	-906.642	0.008123
Clostridium_butyricum_35	1	43	141	172.7327	0.005386
Clostridium_butyricum_35	2	1	6	-5.78975	0.019389
Bifidobacterium_animalis_36	1	1	25	-50.8092	0.005386
Haemophilus_38	1	30	132	278.5522	0.005386
Haemophilus_38	2	1	12	-24.1688	0.008123
Lactobacillus_40	1	35	62	-25.5163	0.015169
Clostridium_paraputrificum_41	1	57	363	687.7779	0.015169
Corynebacterium_47	1	1	181	123.3441	0.011798
Bacteroides_48	1	36	334	-151.544	0.005386
Lactobacillus_51	1	1	329	-175.345	0.034918
Streptococcus_anginosus_subsp_whileyi_55	1	1	68	-82.4844	0.013764
Bacteroides_58	1	74	384	-272.878	0.015169
Bifidobacterium_63	1	1	343	-564.998	0.008123

Akkermansia_64	1	15	384	-229.221	0.013764
Bacteroides_70	1	1	338	-444.768	0.010182
Roseburia_72	1	368	384	48.52357	0.046267
Actinomyces_73	1	49	184	-102.652	0.046267
Erysipelotrichaceae_74	1	81	384	-210.443	0.044351
Peptostreptococcaceae_75	1	368	384	65.41373	0.005386
Peptostreptococcaceae_75	2	126	134	-16.3938	0.005386
Peptostreptococcaceae_75	3	316	361	-465.417	0.005386
Veillonella_76	1	1	54	-77.2432	0.010182
Corynebacterium_striatum_77	1	64	384	214.516	0.005386
Lactococcus_lactis_78	1	1	16	16.38635	0.034918
Lachnospiraceae_86	1	214	384	301.315	0.032083
Veillonella_87	1	1	39	-36.8104	0.005386
Stenotrophomonas_maltophilia_91	1	1	196	71.06601	0.005386
Tepidimonas_95	1	1	27	48.33798	0.005386
Tepidimonas_95	2	51	76	-25.0645	0.005386
Peptostreptococcus_98	1	1	177	-159.217	0.02082
Coprococcus_99	1	140	384	-187.329	0.039201
Lachnospiraceae_103	1	122	384	234.6748	0.017003
Clostridium_sensu_stricto_1_110	1	119	136	8.357675	0.024171
Veillonella_115	1	1	16	-13.6335	0.018018
Enhydrobacter_119	1	1	33	33.19319	0.005386
Gemella_120	1	110	127	19.03825	0.005386
Gemella_120	2	1	32	-73.8688	0.005386
Bifidobacteriaceae_122	1	1	134	-119.583	0.029147
Bilophila_wadsworthia_3_1_6_123	1	1	384	-598.592	0.005386
Clostridium_sensu_stricto_1_124	1	1	163	114.8586	0.010182
Comamonadaceae_130	1	1	27	28.35392	0.005386
Sutterella_134	1	1	381	-305.891	0.015169
Atopobium_138	1	126	174	-9.43428	0.03052
Acinetobacter_143	1	1	27	28.58536	0.005386
Bifidobacterium_146	1	1	166	-176.161	0.010182
Akkermansia_153	1	51	324	-114.827	0.025766
Escherichia_Shigella_154	1	1	29	-76.9747	0.005386
Burkholderia_156	1	1	27	19.42555	0.005386
Acinetobacter_157	1	1	181	111.947	0.008123
Collinsella_tanakaei_163	1	30	138	-40.375	0.018018
Fingoldia_164	1	1	134	117.3084	0.005386
Dorea_170	1	369	384	49.74354	0.013764
Dorea_170	2	267	363	-847.459	0.013764
Eggerthella_178	1	319	384	114.4042	0.022711
Eggerthella_178	2	1	157	-179.476	0.015169
Ruminococcus_sp_14531_180	1	16	212	67.83765	0.029147

Enterococcus_186	1	1	44	31.95333	0.037737
Lactococcus_189	1	1	23	30.36847	0.005386
Lactobacillus_delbrueckii_subsp_bulgaricus_190	1	152	384	-158.129	0.044351
Streptococcus_192	1	1	20	-37.9859	0.005386
Bacteroides_194	1	133	384	-203.277	0.013764
Akkermansia_196	1	62	371	-125.98	0.015169
Klebsiella_200	1	15	131	272.8402	0.005386
Veillonella_sp_DNF00869_201	1	78	126	39.56797	0.005386
Veillonella_sp_DNF00869_201	2	1	21	-20.473	0.005386
Bifidobacterium_animalis_202	1	1	74	-39.9736	0.033939
Parabacteroides_distasonis_207	1	1	245	-105.104	0.011798
Bacteroides_210	1	10	384	-416.578	0.010182
Alistipes_215	1	37	342	-190.377	0.018018
Bifidobacterium_223	1	1	177	-152.024	0.018018
Collinsella_227	1	20	162	-52.3616	0.03315
Bifidobacterium_228	1	1	279	-241.208	0.011798
Bifidobacterium_233	1	1	175	-109.88	0.015169
Bacteroides_236	1	117	384	-197.459	0.010182
Collinsella_237	1	1	334	-871.444	0.011798
Actinobaculum_schaalii_FB123_CNA_2_239	1	25	139	-99.1253	0.008123
Streptococcus_241	1	1	74	-33.4764	0.008123
Bacteroides_247	1	1	286	-192.724	0.008123
Bacteroides_249	1	262	384	-79.917	0.018018
Bifidobacterium_251	1	1	264	-263.264	0.025766
Alistipes_255	1	132	384	-145.27	0.039201
ratAN060301C_268	1	1	328	-257.643	0.005386
Leuconostoc_275	1	1	24	19.91756	0.005386
Leuconostoc_275	2	118	130	-6.31147	0.005386
Barnesiella_276	1	96	384	-136.028	0.019389
Ruminococcaceae_277	1	136	384	-123.329	0.039201
Leuconostoc_mesenteroides_282	1	1	25	23.34859	0.005386
Lactobacillales_284	1	38	301	-279.476	0.02082
Ruminococcaceae_287	1	197	384	-119.482	0.03315
Bifidobacteriaceae_290	1	1	17	-9.20944	0.026888
Bacteroides_fragilis_CL03T00C08_291	1	23	382	-264.105	0.029147
Actinomyces_292	1	169	241	28.24573	0.034918
Pasteurella_pneumotropica_301	1	1	13	-9.24825	0.013764
Bifidobacteriaceae_320	1	83	190	-141.005	0.011798
Escherichia_Shigella_331	1	1	9	-7.23429	0.008123
Gardnerella_332	1	1	384	-326.213	0.011798
Bifidobacterium_333	1	1	26	-41.6445	0.005386
Dermabacter_335	1	1	171	-64.4005	0.039201
Bifidobacterium_336	1	49	212	-51.6362	0.026888

Streptococcus_338	1	1	25	10.74031	0.019389
Clostridium_sensu_stricto_1_341	1	1	118	45.1547	0.033939
Bifidobacteriaceae_344	1	1	10	-7.4067	0.005386
Bifidobacteriaceae_344	2	117	137	-18.5781	0.017003
Clostridium_difficile_630_345	1	1	78	-28.104	0.018018
Bifidobacterium_347	1	42	384	-225.723	0.018018
Clostridium_sensu_stricto_1_355	1	30	125	68.66705	0.005386
Dialister_365	1	63	384	-114.24	0.010182
Bifidobacterium_368	1	3	221	-184.418	0.005386
Lactobacillus_371	1	1	95	-27.6677	0.033939
Streptococcus_galloyticus_subsp_macedonicus_386	1	1	181	-58.547	0.011798
Lactobacillus_390	1	79	220	-117.975	0.037737
Peptostreptococcaceae_396	1	17	155	-62.6028	0.044351
Bacteroides_400	1	121	384	-141.986	0.011798
Actinomyces_sp_oral_clone_DR002_406	1	18	114	109.3833	0.024171
Corynebacterium_propinquum_408	1	1	58	-39.9928	0.028154
Lachnospiraceae_419	1	205	384	162.215	0.044351
Staphylococcus_427	1	1	53	-24.6208	0.039201
Corynebacterium_freneyi_429	1	1	71	18.78578	0.038863
Coprobacter_443	1	54	205	-38.3623	0.02082
Bacteroides_445	1	1	384	-227.248	0.010182
Staphylococcaceae_452	1	1	57	28.1432	0.038863
Streptococcus_galloyticus_subsp_macedonicus_464	1	1	20	-13.5917	0.008123
Escherichia_Shigella_469	1	1	164	-58.4555	0.028154
Bacilli_471	1	1	65	-43.2404	0.005386
Bifidobacterium_breve_473	1	1	60	-42.2703	0.008123
Bacteroides_482	1	213	384	-88.5204	0.017003
Plesiomonas_485	1	39	198	88.8588	0.015169
Bacteroides_493	1	56	227	-62.3357	0.026888
Veillonella_502	1	1	6	-2.77022	0.005386
Corynebacteriaceae_504	1	1	13	7.106628	0.010182
Corynebacteriaceae_504	2	38	122	-65.9096	0.017003
Staphylococcus_507	1	1	18	-11.9003	0.048666
Streptococcus_521	1	1	214	-79.5663	0.011798
Bacteroides_535	1	84	384	-88.7888	0.019389
Staphylococcus_565	1	1	22	-22.5613	0.005386
Ruminococcus_sp_CE2_596	1	105	384	-134.437	0.03052
Bacteroides_603	1	160	363	-68.0615	0.019389
Lachnospiraceae_614	1	136	384	-81.6236	0.046267
Lactobacillus_623	1	1	129	-20.4557	0.047473
Bifidobacterium_animalis_631	1	255	384	45.60759	0.03315
Bacteroides_645	1	1	149	38.29246	0.026888
Campylobacter_646	1	1	150	37.55262	0.033939

Bacteroides_657	1	1	285	-84.5281	0.005386
Bacteroides_658	1	63	384	-107.687	0.024171
Clostridium_sensu_stricto_1_673	1	60	188	26.37826	0.029147

Differential abundance testing by smoothing spline analysis of variance (SS-ANOVA) was executed to test in which specific intervals significant differences in OTUs existed between the groups, adjusted for mode of delivery and breastfeeding. The OTUs are arranged in descending order based on their relative abundance in this dataset. A positive Area value indicates that the abundance of a specific OTU is higher in the amoxicillin + cefotaxime regimen, while a negative area value indicates that the abundance of that OTU is higher in the controls. To correct for multiple testing, the Benjamini-Hochberg method was applied, and the adjusted p-values (p.adj) are shown. Int.no = interval number. In some cases, multiple OTUs of individual bacterial species were identified, so OTUs are referred to by their taxonomical annotations and a rank number (shown in parentheses), which is based on the abundance of each given OTU in the overall dataset.

The reviewer is a bit puzzled about the fact that whereas in the first version of the manuscript significant differences in % later AB were found in one of the groups and when they went back to review their data 4 more cases emerged in this group resulting in non-significant differences. How did the authors monitor their data from the parent reports?

We thank the Reviewer for this important query. For data collection we used the Research Manager software. The questionnaires at baseline about perinatal characteristics were filled in by our research nurses and research physicians. Questionnaires about health characteristics at age 1, 4, 6, 8, 10 and 12 months were filled in by parents online. Once completed, these questionnaires were saved and could not be edited by the parents.

The only exception was the questionnaire on medication use throughout the first year of life, which was editable. In the medication questionnaire parents had the possibility to add information on new medication and input the end date of previously reported medication (because at the time of filling out the questionnaire a treatment could still be ongoing). The first question of the medication questionnaire was “has your child received any medication after discharge from the hospital”. Unfortunately, in some cases the medication data was overwritten, due to some parents changing the answer to this first question from “yes” to “no”, interpreting the question as “has your child received any new many medication since the previous questionnaire”. Thankfully, the Research Manager software saves and tracks all changes made in each questionnaire, so the data was not lost. After a query of conflicting data in two consecutive questionnaires, the team checked all medication questionnaires from all participants for inconsistencies, and corrected where necessary. This resulted in a correction of our analyses as well. The medication questionnaire was the only one in which data could be edited by parents, so errors could occur in this way. We are confident that all other parent reported data were correctly extracted. We apologize for this confusion, and hope this explanation is found satisfactory addressed.

Finally if the reviewer has read the revised version correctly about half of the infants in the groups received AB for 1-4 days and the other half 5-7 days? It is remarkable that in 2 groups the median is < 3 and in the third group it is 4.5 suggesting that in the first 2 groups most infants received 2-3 days AB and in the third group most received >4 days or is the distribution skewed and did those who received longer AB mostly 7 days. My question is that in several studies prolonged i.e. >4 days AB is associated with more adverse health outcomes and microbiota outcomes which were not found in this study. I think this remarkable finding and the possible reasons for not finding this effect in this study whereas it is found in most other studies should be discussed in more detail in the discussion.

We thank the Reviewer for this comment. The duration of antibiotic treatment is not normally distributed, hence we report the medians and interquartile ranges. Indeed, the co-amoxiclav + gentamicin group received relatively more often antibiotics for >4 days compared to the other two groups. The data is not extremely skewed. In our study, we found a modest effect for antibiotic treatment duration in association with microbiota composition (R^2 0.3%, or 0.4% when categorized into 1-4 versus >4 days of treatment; this information now added at lines 159-161 for full transparency), but the effect was small when compared to the initiation of antibiotics in the first place (R^2 0.9%). In our revised version we have more clearly underlined the importance of this finding in the discussion at lines 379-381.

Reviewer #3

I have gone through the revised and resubmitted version of the manuscript by Ms Reyman and colleagues. The quality of the manuscript has improved.

We are glad the Reviewer finds the quality of the manuscript improved.

Specific comments

Still, there is one important crucial issue that should be more emphasized in the manuscript. Most of the significant differences are found between the RCT cohort (infants receiving antibiotics) and the control cohort (infants not receiving antibiotics). I understand that it was difficult to include non-antibiotic infants in the RCT due to ethical reasons. However, it still means that there is a high risk of systematic bias. We know from other studies that the gut microbiome differs between different cohorts. In the present study the two cohorts included infants from the same hospitals but it seems, however, that the recruitment was separated by time, and the gut microbiome composition is known to fluctuate over time.

We thank the Reviewer for this remark and agree with these potential confounding issues. We have better underlined the differences between the two cohorts by adding the information requested by the Reviewer in the specific suggestions below. Hopefully, having pre-antibiotic exposure data from both cohorts, showing limited and non-significant differences between the cohorts, plus our adapted post hoc analyses correcting for potentially important co-drivers of microbial development, may help to convince the Reviewer that the results presented are reliable.

Suggestions

Add the years when the control cohort was including infants in the result section.

The control infants were born between 19 December 2012 and 2 November 2014. We have now added this information in the results section at line 83.

Add the years when the RCT and the control cohort, respectively, was including infants in the abstract.

We have now added this information in the abstract at lines 26 and 30.

Discuss the risk of systematic bias as described above in the discussion section.

We have further emphasized and expanded the possibility of bias in the discussion section at lines 407-408 and 412-415.

Reviewer #4

In general, they addressed all the issues raised by Reviewer 1.

We thank the Reviewer for his/her positive assessment.

Specific comments

The only part that I still believe needs some attention is the following point regarding breastfeeding. I totally agree with reviewer 1 regarding the inclusion of breastfeeding in the analysis, especially since bifidobacterium was found to be differential. I think the answer by the authors is good, but I would like to see that answer in the revised paper (for example in the supplemental information). Currently, it's only part of the response to the reviewers.

We thank the Reviewer for this suggestion and now better understand the issue raised by Reviewer 1.. We have therefore now added the described and new information in the manuscript and Supplementary Information (also see answer to Reviewer 1) and highlighted both the findings in the overarching analysis (lines 136-145) and the fitTimeSeries (OTU-specific) analysis (lines 207-219 and eTables 8 and 9).

In addition, the authors claim that they only examined features that differentiated across the groups, but it seems like breastfeeding was indeed different, for example, at 4 months of age, so I'm not 100% sure why it was not included.

We agree, and as stated above, we now more clearly underline the already previously added variable of breastfeeding duration in the overarching multivariate analyses (lines 136-145) and present the additional longitudinal fitTimeSeries results including duration of breastfeeding in the manuscript (lines 207-219 and eTables 8 and 9) to ensure both the effect of breastfeeding on community level as well as the limited compensation for antibiotic-induced effects (see Reviewer 1) are highlighted.

In addition, it took me a really long time to understand the type of data the paper is based on. I searched the abstract, end of introduction, and only in the methods section I realized that the vast majority of analysis is based on PCR of known AMR. Is that so? In any case, this information should be made clearer MUCH earlier in the paper. As a reader, I assumed this is metagenomic sequencing, but when I got to the region discussing the few samples that underwent further investigation with mgx sequencing, I realized something was missing.

We regret it was unclear early on in the manuscript what type of data the results are based on. We have clarified this in more detail in the revised manuscript at lines 68-72.

It might be just language, but I found this statement to be confusing. I imagine/hope that all data were deposited to NCBI, and not only the subset that supported these findings, right? :) “Sequence data that support the findings of this study have been deposited in the NCBI Sequence Read Archive (SRA) database with BioProject IDs PRJNA481243, PRJNA524461 and PRJNA555020.”

Indeed, all the data were deposited to NCBI, and we apologize for the confusion.

References

1. Reyman, M. *et al.* Impact of delivery mode-associated gut microbiota dynamics on health in the first year of life. *Nat. Commun.* **10**, 4997 (2019).

REVIEWERS' COMMENTS

Reviewer #2 (Remarks to the Author):

The authors have addressed my remaining comments sufficiently and I have no further comments.